# When Does Sparsity Mitigate the Curse of Depth in LLMs

Dilxat Muhtar [* 1 2 3]  Xinyuan Song [* 4]  Sebastian Pokutta [5 6]  Max Zimmer [5 6]  Nico Pelleriti [5 6]
Thomas Hofmann [7]  Shiwei Liu [1 2 3]

## Abstract

Recent work has demonstrated the *curse of depth* in large language models (LLMs), where later layers contribute less to learning and representation than earlier layers. Such under-utilization is linked to the accumulated growth of variance in Pre-Layer Normalization, which can push deep blocks toward near-identity behavior. In this paper, we provide evidence that *sparsity*-like mechanisms can dampen variance propagation and are associated with ***improved depth utilization*** Our investigation covers two sources of sparsity: (i) implicit sparsity, which emerges from training and data conditions, including weight sparsity induced by weight decay and attention sparsity induced by long-context inputs; and (ii) explicit sparsity, which is enforced by architectural design, including key/value-sharing in Grouped-Query Attention and expert-activation sparsity in Mixture-of-Experts. Our claim is thoroughly supported by controlled depth-scaling experiments and targeted layer effectiveness interventions. Across settings, we observe a consistent relationship: mechanisms with reduced effective interaction density tend to exhibit lower output variance and better layer differentiation. We eventually distill our findings into a practical rule-of-thumb recipe for training depth-effective LLMs, yielding a notable ***4.6 accuracy improvement*** on downstream tasks. Our results suggest that sparsity-like design choices are an important and previously under-emphasized factor in effective depth scaling for LLMs. Code is available at https://github.com/pUmpKin-Co/SparsityAndCoD.

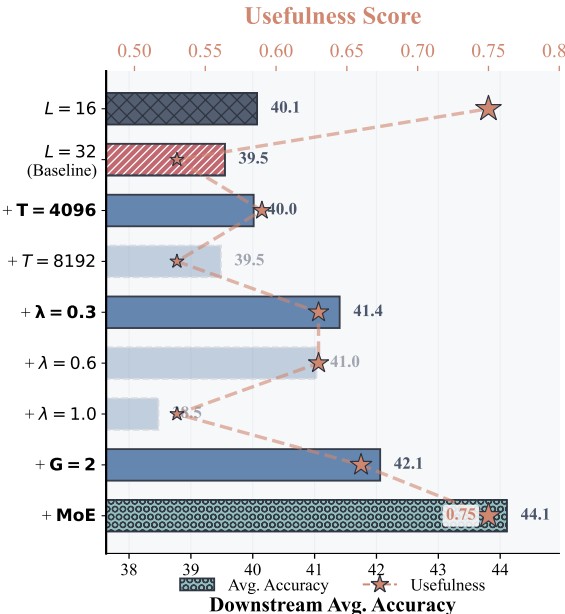

*Figure 1.* Progressive performance gains when scaling the 1.2B model to depth $L = 32$ via stacking with various "sparsity" modules.

## 1. Introduction

Although LLMs exhibit remarkable capabilities, growing evidence shows that later Transformer layers are frequently under-utilized, contributing little to final performance (Gromov et al., 2024; Men et al., 2025; Csordás et al., 2025). For instance, recent studies demonstrate that skipping layers in LLMs incurs negligible performance degradation (Lad et al., 2024; Yang et al., 2024; Ma et al., 2024a). This phenomenon reveals layer redundancy that, while enabling model compression through layer pruning (Li et al., 2025a; Dumitru et al., 2024; Yin et al., 2023; Zheng et al., 2025), indicates inefficient utilization of training resources (Du et al., 2024; Kapl et al., 2025; Kamigaito et al., 2025).

Sun et al. (2025) have recently summarized this phenomenon as the *curse of depth (CoD)* and identified variance propagation as a key underlying cause of this ineffectiveness. In widely adopted Pre-Layer Normalization (Pre-LN) architectures (Xiong et al., 2020; Kan et al., 2025; Wang et al., 2022), output variance tends to grow sub-exponentially with model depth (Sun et al., 2025; Takase et al., 2023). As variance accumulates, the magnitude of the residual stream dwarfs the updates provided by individual layers, causing deep layers to become functionally ineffective as their Jaco-

[1]Max Planck Institute for Intelligent Systems [2]ELLIS Institute Tübingen [3]Tübingen AI Center [4]Emory University [5]Zuse Institute Berlin [6]Technical University of Berlin [7]ETH Zürich. Correspondence to: Shiwei Liu <sliu@tue.ellis.eu>, Dilxat Muhtar <pumpkindilxat@gmail.com>.

*Proceedings of the $43^{rd}$ International Conference on Machine Learning*, Seoul, South Korea. PMLR 306, 2026. Copyright 2026 by the author(s).

bians approach the identity. Consequently, the community has largely focused on explicit variance control to mitigate this accumulation, such as Scaled Initialization (Zhang et al., 2019; Luther and Seung, 2019; Takase et al., 2023), LayerNorm Scaling (Sun et al., 2025), advanced residual connections (Zhu et al., 2025; Xie et al., 2025), and alternative normalization like Mix-LN (Li et al., 2024; Cai et al., 2025; Ding et al., 2021; Wang et al., 2024).

In parallel, a second trend has emerged in modern LLMs: the widespread adoption of *sparse computation*. We use the term sparsity broadly here to include mechanisms that reduce effective interaction density, parameter participation, activation mass, or independent computational paths. Contemporary architectures increasingly incorporate sparsity at multiple levels: Mixture of Experts (MoE) activates only parameter subsets (Liu et al., 2025; Yang et al., 2025), Grouped Query Attention (GQA) reduces attention density (Ainslie et al., 2023; Shazeer, 2019), and extended sequence lengths naturally induce sparse attention patterns (Yuan et al., 2025; Xiao et al., 2023). While these innovations are typically justified by efficiency gains, their impact on variance propagation dynamics remains poorly understood. Intriguingly, these two approaches may be more deeply connected than previously recognized. Prior study has documented "signal collapse" in certain sparse networks, where variance diminishes as connection density decreases (Dey et al., 2024; 2025), suggesting that sparsity might inherently regulate variance. This observation raises a compelling question: *could sparsity—whether explicitly enforced through architecture or implicitly induced through training—serve as an intrinsic mechanism for mitigating the CoD by regulating variance propagation?*

In this work, we provide both theoretical and empirical evidence that sparsity serves as an intrinsic variance regulator that mitigates the CoD. We begin by characterizing the CoD through controlled experiments. We train models from scratch across varying depths (12 to 32 layers) while holding all other hyperparameters constant. To quantify layer effectiveness, we introduce three metrics: (1) Causal Score measures how much removing a layer disrupts subsequent layer representations; (2) Permutation Score quantifies layer interchangeability; and (3) Usefulness Score evaluates each layer's contribution to final performance. We then provide a formalization of how sparsity counteracts depth-induced variance accumulation. We analyze two distinct paradigms: *implicit sparsity*, i.e., weight sparsity induced through weight decay and attention sparsity caused by long-context input; and *explicit sparsity*, i.e., enforced via GQA-style key/value sharing and sparse MoE routing. We systematically compare variance propagation and layer effectiveness across: (a) weight decay strengths, (b) sequence length scaling, (c) different GQA configurations (varying number of query groups), and (d) two MoE model scales (2B and 7B

parameters) alongside their dense counterparts. Across all settings, we observe a consistent pattern: *increased sparsity correlates with reduced output variance and improved layer effectiveness.* Finally, we distill our findings into a practical rule of thumb for training depth-effective LLMs. By integrating complementary sparsity mechanisms, we train a 32-layer, 1.2B-parameter model that achieves stronger performance and improved layer effectiveness compared with a naively trained 32-layer baseline (Figure 1).

The main contributions are summarized as follows:

- We leverage three metrics, i.e., Causal Score, Permutation Score, and Usefulness Score, to quantify layer effectiveness. Using controlled depth-scaling experiments, we show that deeper models exhibit degraded layer utilization, providing empirical evidence of the *curse of depth*.

- We show that both implicit sparsity (e.g., weight decay and long-context inputs) and explicit sparsity (e.g., Mixture of Experts and Grouped Query Attention) mitigate residual-stream variance propagation, consistently reducing variance accumulation and improving layer effectiveness.

- We distill our findings into a simple rule-of-thumb for training depth-effective LLMs. combining complementary sparsity mechanisms yields a notable **4.6 accuracy gain** on downstream tasks.

## 2. Variance Propagation and Curse of Depth

In a Pre-LN (Xiong et al., 2020; Wang et al., 2024) Transformer block at layer $\ell$, the forward pass applies layer normalization before the transformation:

$$\mathbf{x}_{\ell+1} = \mathbf{x}_\ell + \mathcal{F}(\text{LN}(\mathbf{x}_\ell)), \tag{2.1}$$

where $\mathbf{x}_\ell \in \mathbb{R}^d$ is the input to layer $\ell$, $\mathcal{F}(\cdot)$ denotes either a Multi-Head Attention (MHA) or FFN module, and $\text{LN}(\cdot)$ is layer normalization.

**Lemma 1** (gradient converge to identity (Sun et al., 2025) Theorem 3.3). *For a Pre-LN Transformer with $L$ layers using Equations* (2.1)*, assuming that the input vectors, intermediate vectors, and parameter follow independent zero-mean Gaussian distributions, and that $\sigma^2_{x_\ell}$ grows exponentially, then the partial derivative $\frac{\partial y_L}{\partial x_1}$ can be written as:*

$$\frac{\partial y_L}{\partial x_1} = \prod_{\ell=1}^{L-1} \left( \frac{\partial y_\ell}{\partial x'_\ell} \cdot \frac{\partial x'_\ell}{\partial x_\ell} \right). \tag{2.2}$$

*We define the intermediate state $x'_\ell \in \mathbb{R}^d$ as the post-attention, pre-FFN residual state: $x'_\ell := x_\ell + \text{Attn}_\ell(x_\ell)$,*

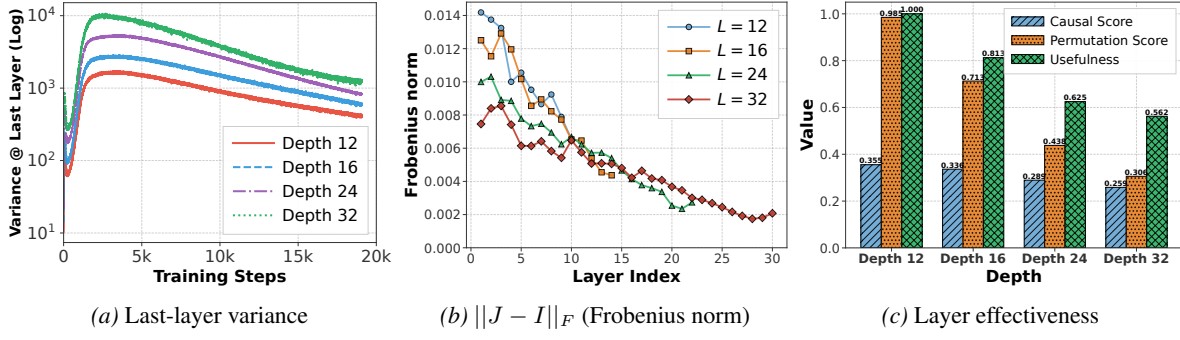

*(a) Last-layer variance*      *(b) $||J - I||_F$ (Frobenius norm)*      *(c) Layer effectiveness*

*Figure 2.* **(a)** Last-layer variance increases with depth under controlled width (note: total parameter count varies across depths). **(b)** Jacobian Frobenius norm $||J - I||_F$. **(d)** Layers in deeper models exhibit lower effectiveness (usefulness and causality) and higher redundancy (permutation scores).

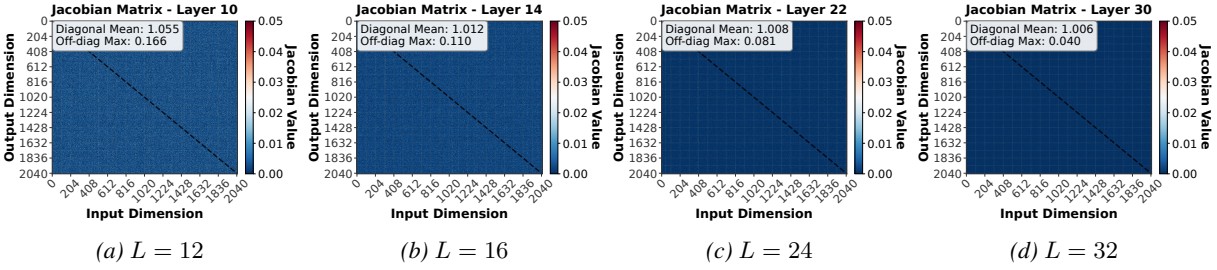

*(a) $L = 12$*      *(b) $L = 16$*      *(c) $L = 24$*      *(d) $L = 32$*

*Figure 3.* Jacobian matrices at layers $\{10, 14, 22, 30\}$ for $L \in \{12, 16, 24, 32\}$.

and the block output as $y_\ell := x'_\ell + \text{FFN}_\ell(x'_\ell)$. *The Euclidean norm of the upper bound for Equation (2.2) is given as follows:*

$$\sigma_{x_\ell}^2 \sim \exp(\ell), \quad \left\| \frac{\partial y_L}{\partial x_1} \right\|_2 \leq M, \quad (2.3)$$

From Lemma 1, under conditions of exponential variance growth, the Jacobian norm $\left\| \frac{\partial y_L}{\partial x_1} \right\|_2$ remains uniformly bounded by $M$ as $L \to \infty$, with $M$ denoting the asymptotic limit to which the gradient norm converges. Therefore, depth alone does not necessarily cause gradient instability: the bounded Jacobian norm suggests that the Transformer can remain stable in norm as depth increases. Consequently, when $L$ is very large, deeper layer transformations approach **identity mappings** from $x_\ell$ to $y_\ell$, restricting expressivity and the model's ability to learn nontrivial mappings.

**Experimental Setup** To verify the accumulation of variance with model depth and examplify the CoD, we conduct controlled experiments using Pre-LN architectures. To isolate depth effects, we vary only the number of layers from 12 to 32, keeping all other architectural and training configurations fixed across experiments. For each configuration, we perform a learning rate sweep and report results for the best-performing setting on validation data. We track last-layer output variance during training and define three metrics to quantify layer effectiveness. To verify Jacobian convergence to identity, we measure each layer's Frobenius deviation $||J - I||_F$. Details are provided in Section C.1.

**Last-Layer Variance** For hidden states $\mathbf{h}_L \in \mathbb{R}^{n \times d}$, we compute variance across dimensions, averaged over tokens:

$$\text{Var}(L) = \frac{1}{n} \sum_{i=1}^{n} \text{Var}(\mathbf{h}_{L,i,:}) = \frac{1}{nd} \sum_{i=1}^{n} \sum_{j=1}^{d} \left( h_{L,i,j} - \bar{h}_{L,i} \right)^2, \quad (2.4)$$

where $\bar{h}_{L,i} = \frac{1}{d} \sum_j h_{L,i,j}$ is the per-token mean. High variance indicates signal accumulation across depth, causing layer gradient to become negligible (Sun et al., 2025)

**Causal Score** The causal score measures how much each layer influences the computations of all subsequent layers (Csordás et al., 2025). For a model with $N$ layers and hidden states $\mathbf{h}_\ell$ at layer $\ell$, we define the causal effect of layer $s$ on layer $\ell > s$ as:

$$C(s, \ell) = \frac{\|(\mathbf{h}_{\ell+1} - \mathbf{h}_\ell) - (\bar{\mathbf{h}}_{\ell+1} - \bar{\mathbf{h}}_\ell)\|_2}{\|\mathbf{h}_{\ell+1} - \mathbf{h}_\ell\|_2}, \quad (2.5)$$

where $\mathbf{h}_\ell$ denotes hidden states in the baseline model, and $\bar{\mathbf{h}}_\ell$ denotes hidden states when layer $s$ is skipped. The global causal score aggregates these effects across all layer pairs:

$$S_{\text{causal}} = \frac{1}{\sqrt{N}} \cdot \frac{1}{N} \sum_{s=0}^{N-1} \left[ \frac{1}{N-s-1} \sum_{\ell=s+1}^{N-1} C(s, \ell) \right], \quad (2.6)$$

where $1/\sqrt{N}$ normalizes for model depth. Higher causal scores indicate critical layers whose removal affects subsequent layers, while lower scores suggest minimal impact and potential redundancy.

**Permutation Score**   The permutation score quantifies layer specialization by measuring performance degradation when layer positions are swapped (Kapl et al., 2025). For layers $\ell_1$ and $\ell_2$, the pairwise permutation score is:

$$P(\ell_1, \ell_2) = \frac{|\mathcal{L}(M) - \mathcal{L}(M_{\text{swap}(\ell_1, \ell_2)})|}{|\mathcal{L}(M)|}, \qquad (2.7)$$

where $\mathcal{L}(M)$ is the baseline loss and $\mathcal{L}(M_{\text{swap}(\ell_1,\ell_2)})$ is the loss after swapping. The global permutation score averages over all possible layer pairs:

$$S_{\text{permutation}} = \frac{2}{N(N-1)} \sum_{\ell_1=0}^{N-1} \sum_{\ell_2=\ell_1+1}^{N-1} P(\ell_1, \ell_2). \qquad (2.8)$$

Higher scores indicate that layers are less interchangeable, while scores near zero suggest redundancy.

**Usefulness Score**   The usefulness score quantifies each layer's contribution through linear approximation (Kapl et al., 2025; Csordás et al., 2025; Sun et al., 2025). This approach measures the degree of nonlinearity each layer contributes, which is a fundamental indicator of computation beyond linear mappings.

For each layer $\ell$, we collect input-output pairs $\{(\mathbf{x}_i, \mathbf{y}_i)\}_{i=1}^N$ and fit an optimal linear approximation via least-squares. We then measure performance degradation when replacing layer $\ell$ with this linear transformation:

$$\Delta\mathcal{L}_\ell = \mathcal{L}(M_\ell^{\text{linear}}) - \mathcal{L}(M). \qquad (2.9)$$

The global usefulness score measures the fraction of layers with significant ($> \alpha$) performance impact:

$$S_{\text{useful}} = \frac{1}{L} \sum_{\ell=1}^{L} \mathbb{1}\left[\frac{\mathcal{L}(M_\ell^{\text{linear}})}{\mathcal{L}(M)} > 1 + \alpha\right], \qquad (2.10)$$

where we set $\alpha = 0.1$. This quantifies the model's *effective nonlinear depth*—the fraction of layers performing meaningful nonlinear transformations. Higher scores indicate efficient depth utilization; lower scores reveal redundancy.

**Main Observation**   We observe a consistent empirical association from variance accumulation to diminished layer effectiveness. Figure 2a shows that variance grows substantially with depth, aligning with (Sun et al., 2025). This variance accumulation is accompanied by Jacobian matrices toward identity mapping: Figure 2b reveals that the Frobenius norm $||J - I||_F$ decreases with depth, while Figure 3 shows increasingly diagonal-dominant Jacobian patterns in deeper models, supporting Lemma 1. Consequently, layer effectiveness deteriorates: Figure 2c demonstrates that all three score progressively decline, with Usefulness dropping from 0.75 ($L = 12$) to 0.53 ($L = 32$). While $L = 32$ achieves better performance (Table 1) with more effective

*Table 1.* Evaluation results for models with different depths. E/W represents effective layers (E) versus low-effectiveness layers (W) as measured by the usefulness score (Section 2).

| Depth$L$ | # of Param. | E/W | PPL↓ | MMLU | ARC-C | ARC-E | Hellaswag |
|---|---|---|---|---|---|---|---|
| L=12 | 900M | 12/0 | 13.72 | 24.20 | 30.89 | 58.96 | 47.24 |
| L=16 | 1.2B | 13/3 | 13.65 | 25.20 | 31.72 | 60.45 | 47.24 |
| L=24 | 1.7B | 15/9 | 13.53 | 26.39 | 32.59 | 61.24 | 47.50 |
| L=32 | 2.3B | 18/14 | **13.47** | **27.52** | **34.22** | **61.74** | **48.63** |

layers (18 vs. 12 for $L = 12$), it exhibits severe inefficiency: using $2.56\times$ more parameters while having 14 low-effectiveness layers, exemplifying CoD where most layers contribute minimally despite consuming substantial training compute. Additional analyses are provided in Sections A.1 to A.4.

## 3. Sparsity as Variance Regularizer

Having established that variance propagation is a contributor to CoD, we now investigate sparsity as a mechanism to control variance accumulation.

### 3.1. Theoretical Analysis

Sparsity acts as a variance regularizer in residual stacks by attenuating the energy passed to each layer update. The following lemma quantifies this effect: the per-layer variance gain scales as $\alpha_\ell \rho_\ell$, so smaller mask density $\rho_\ell$ yields slower variance growth with depth. To isolate the effect of reduced interaction density, we first analyze a simplified residual recursion. This result should be read as an abstract proxy model rather than a faithful description of Transformer dynamics.

**Theorem 1** (Sparsity reduces variance propagation in residual-depth)**.** *Let* $\{r_\ell\}_{\ell=0}^L$ *follow the residual-depth recursion*

$$r_{\ell+1} = r_\ell + u_\ell, \quad u_\ell := W_\ell\big(D_\ell r_\ell\big), \qquad \ell = 0, \ldots, L-1, \qquad (3.1)$$

*where* $r_\ell \in \mathbb{R}^d$, $D_\ell \in \mathbb{R}^{d\times d}$ *is a diagonal 0-1 mask, and* $W_\ell \in \mathbb{R}^{d\times d}$ *is a random linear map. Assume that for each* $\ell$, $W_\ell$ *is independent of* $(r_\ell, D_\ell)$ *and satisfies the second-moment bound and that the mask satisfies the density bound*

$$\mathbb{E}\left[\|W_\ell x\|_2^2\right] \leq \alpha_\ell \|x\|_2^2, \quad \mathbb{E}\left[\|D_\ell x\|_2^2\right] \leq \rho_\ell \|x\|_2^2 \quad (3.2)$$

*for all* $x \in \mathbb{R}^d$, *and for some* $\rho_\ell \in [0, 1]$. *If* $\alpha_\ell \leq \alpha$ *and* $\rho_\ell \leq \rho < 1$ *for all* $\ell$, *then the residual variance satisfies*

$$\text{Var}(r_L) \leq \text{Var}(r_0) \prod_{\ell=0}^{L-1}\left(1 + \sqrt{\alpha_\ell \rho_\ell}\right)^2 = O\left(1 + \sqrt{\alpha\rho}\right)^{2L}. \qquad (3.3)$$

The proof is provided in B.1. Theorem 1 shows that the variance bound depends on sparsity only through $\rho_\ell$: smaller $\rho_\ell$ (sparser $D_\ell$) yields a smaller per-layer factor $(1 + \sqrt{\alpha_\ell \rho_\ell})^2$, and therefore a smaller upper bound on $\text{Var}(r_L)$. Hence, sparsity (captured by $\rho_\ell$) directly controls variance: smaller

$\rho_\ell$ yields a smaller bound on $\mathrm{Var}(r_L)$ across depth. We observe empirically that $\rho_\ell$ can also be mitigated through training-induced sparsity patterns. It is important to note, however, that the theoretical result in Theorem 1 relies on the assumption that the weight $W_\ell$ is independent of the sparsity masking variables $(r_\ell, D_\ell)$. When sparsity is induced during training, the weights and sparsity patterns become inherently coupled, meaning this strict independence no longer holds. Nevertheless, Theorem 1 provides a useful conceptual approximation for understanding how emergent sparsity restrains variance accumulation in practice.

### 3.2. Implicit and Explicit Sparsity

Motivated by our theoretical analysis, we identify specific sparsity dimensions for experimental investigation. We categorize sparsity into implicit sparsity and explicit sparsity.

#### 3.2.1. IMPLICIT SPARSITY

Implicit sparsity refers to sparsity induced dynamically during training. This includes sparsity from regularization (weight decay (Krogh and Hertz, 1991; Loshchilov and Hutter, 2019)), activation functions (ReLU (Glorot et al., 2011; Hayou et al., 2019; Ma et al., 2024b)), and input-dependent mechanisms (dropout (Srivastava et al., 2014), attention patterns (Vaswani, 2017)) that drive parameters or activations toward negligible values (Frankle and Carbin, 2018; Chen et al., 2020; Ma et al., 2025). We investigate two implicit sparsity dimensions: (1) weight decay, and (2) sequence length scaling.

**Weight Decay**   Weight decay applies $L_2$ regularization to model parameters, adding penalty term $\lambda\|W\|_2^2$ to the loss function (Loshchilov and Hutter, 2019). This drives small-magnitude parameters toward zero, inducing sparsity without structural constraints. We quantify the induced sparsity by measuring the fraction of effectively zero parameters. For a trained model with parameter set $\Theta$, we define:

$$\mathrm{Sparsity}(\Theta; \epsilon) = \frac{1}{|\Theta|} \sum_{w \in \Theta} \mathbb{1}[|w| < \epsilon], \qquad (3.4)$$

where $\epsilon$ is a threshold and $\mathbb{1}[\cdot]$ is the indicator function.

Weight decay provides a simple variance-control effect during training. Under the decoupled update, it contracts the contribution of the initialization over time and limits the variance injected by stochastic gradients, which together reduce the variance of downstream layer outputs (Theorem 3). Moreover, in the stable regime $0 < \eta\lambda \leq 1$, increasing $\lambda$ tightens this control, yielding smaller output variance. We therefore interpret weight decay as an optimization-induced implicit regularizer that stabilizes activations by suppressing parameter variance throughout training. The formal statement and proof are deferred to Appendix B.4.

**Sequence Length**   Sequence length scaling induce implicit sparsity in attention mechanisms through positional bias and softmax normalization (Su et al., 2024; Xiao et al., 2023; Zhang et al., 2023). Position embeddings like RoPE (Su et al., 2024) introduce distance-dependent attention decay that the dot product decreases with relative distance. As sequence length $T$ increases, this distance penalty causes attention to concentrate on a subset of positions with favorable relative distances. Softmax normalization over longer sequences produces more peaked distributions, concentrating attention on top-scoring positions while suppressing others toward zero (Xiao et al., 2023; Zhang et al., 2023; Yuan et al., 2025). We quantify attention sparsity by measuring the fraction of near-zero attention weights. For attention weights $\mathbf{A}_{\ell,h} \in \mathbb{R}^{T \times T}$ at layer $\ell$ and head $h$, we compute the sparsity at threshold $\epsilon$ as:

$$\mathrm{Sparsity}_{\ell,h}(\epsilon) = \frac{1}{T^2} \sum_{i=1}^{T} \sum_{j=1}^{T} \mathbb{1}[A_{i,j} < \epsilon], \qquad (3.5)$$

where $\mathbb{1}[\cdot]$ is the indicator function. This measures the percentage of attention weights below $\epsilon$. The global attention sparsity aggregates across all layers and heads:

$$\mathrm{Sparsity}_{\mathrm{global}}(\epsilon) = \frac{1}{L \cdot H} \sum_{\ell=1}^{L} \sum_{h=1}^{H} \mathrm{Sparsity}_{\ell,h}(\epsilon) \qquad (3.6)$$

where $L$ is the number of layers and $H$ is the number of attention heads per layer.

Beyond inducing sparsity, longer sequences average out stochasticity in attention output (Theorem 4; see Appendix B.5 for details): under the uniform-attention approximation, the output behaves like an average over $T$ independent value coordinates, so variance decreases inversely with $T$.

#### 3.2.2. EXPLICIT SPARSITY

Explicit sparsity refers to architectural constraints that hardcode sparsity patterns into the model structure, ensuring that a predetermined fraction of connections or computational paths are absent by design (Ainslie et al., 2023; Dai et al., 2024; Fedus et al., 2022; Abnar et al., 2025). Unlike implicit sparsity that emerges dynamically during training, explicit sparsity is fixed at model initialization through architectural choices. In this study, we examine two prominent forsm of explicit sparsity in LLM design: (1) Grouped Query Attention (GQA), and (2) Mixture of Experts (MoE).

**Grouped Query Attention**   GQA (Ainslie et al., 2023; Shazeer, 2019) reduces attention computation by sharing key-value heads across multiple query heads. In standard multi-head attention with $H$ heads, each head maintains independent query ($\mathbf{Q}_h$), key ($\mathbf{K}_h$), and value ($\mathbf{V}_h$) projections. GQA partitions the $H$ query heads into $G$ groups,

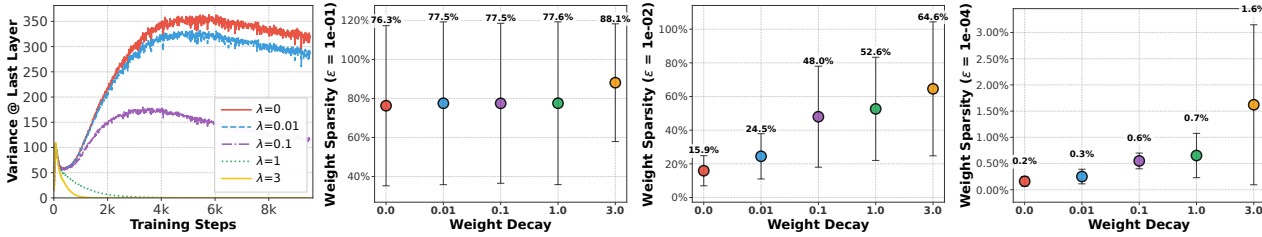

*(a)* Last-layer variance     *(b)* Weight Sparsity ($\epsilon=1\times10^{-1}$) *(c)* Weight Sparsity ($\epsilon=1\times10^{-2}$) *(d)* Weight Sparsity ($\epsilon=1\times10^{-4}$)

*Figure 4.* **(a)** Variance decreases with stronger weight decay. **(b)-(d)** Weight sparsity (fraction of weights $< \epsilon$) increases with weight decay at thresholds $\epsilon \in \{10^{-1}, 10^{-2}, 10^{-4}\}$.

where all queries in a group share the same key-value:

$$\text{Attention}_h(\mathbf{Q}_h, \mathbf{K}_{\lfloor h/G \rfloor}, \mathbf{V}_{\lfloor h/G \rfloor}). \qquad (3.7)$$

This reduces the number of independent key-value computations from $H$ to $H/G$.

Beyond computational efficiency, GQA and MQA can also be interpreted as adding a variance-reduction effect at the final attention output. Under uniform attention weights and independent, zero-mean value rows, each attention head averages over $n$ value vectors, so the per-coordinate variance is on the order of $\sigma_V^2/n$. When the outputs from $G$ heads are further averaged and treated as approximately independent across heads, this aggregation introduces an additional factor of $1/G$. Therefore, in this idealized setting, both GQA and MQA lead to an output variance scaling of approximately $\sigma_V^2/(Gn)$.

**Mixture of Experts** MoE introduces sparsity by replacing dense FFN layers with multiple expert networks, activating only $k$ out of $E$ experts per token (Fedus et al., 2022; Bai et al., 2023; Dai et al., 2024; Bi et al., 2024). Specifically, a gating network routes each token to its top-$k$ experts:

$$\mathbf{h}_{\text{out}} = \sum_{i \in \text{Top-}k(\mathbf{W}_g\mathbf{x})} g_i(\mathbf{x}) \cdot \text{Expert}_i(\mathbf{x}) \qquad (3.8)$$

where $g_i(\mathbf{x})$ are normalized gating weights and $\text{Expert}_i$ are independent FFN networks.

Beyond computational sparsity, Top-$k$ MoE also yields a variance-control effect: because the layer output is an explicit average over the $k$ selected experts, the variability of both the layer output and its local input–output sensitivity (via the Jacobian) is reduced by averaging. Under standard independence and locally-constant routing assumptions, this averaging leads to an approximately $1/k$ reduction in per-coordinate output variance and in Jacobian variance; see Theorem 5 (Appendix B.6) for the formal statement.

## 4. Verification of Variance Dampening

In this section, we validate whether sparsity mitigates variance accumulation and the CoD through end-to-end training.

*Table 2.* Validation perplexity and layer effectiveness scores across weight decay .

| Weight Decay ($\lambda$) | PPL ($\downarrow$) | Effectiveness Score | | |
|---|---|---|---|---|
| | | Causal | Permutation | Usefulness |
| $\lambda = 0$ | 15.63 | 0.25 | 0.41 | 0.75 |
| $\lambda = 0.01$ | 15.20 | 0.26 | 0.42 | 0.75 |
| $\lambda = 0.1$ | **14.83** | **0.31** | **0.52** | **0.81** |
| $\lambda = 1.0$ | 15.55 | 0.20 | 0.50 | 0.69 |
| $\lambda = 3.0$ | 773.42 | 0.03 | 0.07 | 0.63 |

### 4.1. Implicit Sparsity

#### 4.1.1. WEIGHT DECAY

**Experimental Settings** We evaluate the influence of weight decay on model sparsity and variance propagation by training 1.2B-parameter models with varying weight decay coefficients. We employ the AdamW optimizer (Loshchilov and Hutter, 2019) with weight decay values of $\lambda \in \{0, 0.01, 0.1, 1.0, 3.0\}$, while keeping all other hyperparameters fixed. We analyze four key metrics: (1) model parameter sparsity as defined in Section 3.2.1; (2) the evolution of last-layer output variance throughout training; (3) validation perplexity and layer effectiveness scores. Detailed hyperparameters are provided in Section C.2.

**Main Observation** Variance trajectories (Figure 4a) show consistent reduction with stronger weight decay, with last-layer variance decreasing as $\lambda$ increases. At extreme values ($\lambda \in \{1.0, 3.0\}$), variance falls below 25, with $\lambda = 3.0$ driving most weights below $10^{-2}$ (Figure 4c). Weight sparsity increases correspondingly across all thresholds (Figures 4b to 4d), confirming that $L_2$ regularization induces parameter-level sparsity. Performance analysis (Table 2) reveals that within the optimal range ($\lambda \in [0, 0.1]$), both perplexity ($15.63 \rightarrow 14.83$) and usefulness score ($0.75 \rightarrow 0.81$) improve with weight decay, demonstrating that sparsification enhances model quality and depth utilization. However, excessive regularization ($\lambda \geq 1.0$) causes degradation despite continued variance reduction, with $\lambda = 3.0$ leading to collapse (perplexity 773.42). This *over-dampening phenomenon*, where excessive weight decay cripples model capacity despite achieving low variance, demonstrates that variance control must be balanced with model capacity.

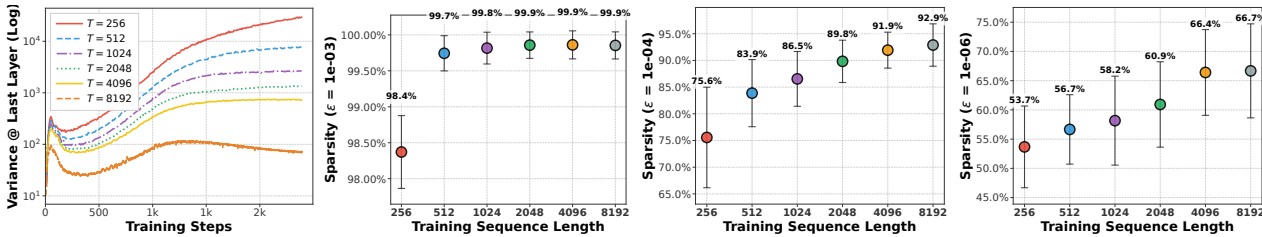

*(a)* Last-layer variance    *(b)* Attn. sparsity ($\epsilon=1 \times 10^{-3}$)    *(c)* Attn. sparsity ($\epsilon=1 \times 10^{-4}$)    *(d)* Attn. sparsity ($\epsilon=1 \times 10^{-6}$)

*Figure 5.* **(a)** Last-layer output variance decreases with longer sequences. **(b)-(d)** Attention sparsity (fraction of weights below threshold $\epsilon$) increases with sequence length across all thresholds. Results averaged over 3 random seeds.



*Figure 6.* Average attention maps from head 0 across all layers for models trained with different sequence lengths (averaged over 100 random held-out samples). Attention weights are power-scaled ($A^{0.2}$) for visualization.

*Table 3.* Validation perplexity and layer effectiveness scores across training sequence lengths.

| Sequence Length ($T$) | PPL ($\downarrow$) | Effectiveness Score | | |
|---|---|---|---|---|
| | | Causal | Permutation | Usefulness |
| $T = 256$ | 18.51 | 0.24 | 0.41 | 0.69 |
| $T = 512$ | 15.71 | 0.25 | 0.45 | 0.75 |
| $T = 1024$ | 14.77 | 0.28 | 0.46 | 0.75 |
| $T = 2048$ | **14.51** | 0.30 | 0.50 | 0.81 |
| $T = 4096$ | 14.52 | **0.31** | **0.81** | 0.81 |
| $T = 8192$ | 16.30 | 0.29 | 0.49 | 0.75 |

*Table 4.* Impact of GQA on model performance and layer-wise effectiveness.

| Group Size ($G$) | PPL ($\downarrow$) | Effectiveness Score | | |
|---|---|---|---|---|
| | | Causal | Permutation | Usefulness |
| $G = 1$ | 14.52 | 0.31 | 0.51 | 0.81 |
| $G = 4$ | 14.50 | 0.31 | 0.57 | **0.87** |
| $G = 16$ | **14.47** | **0.34** | **0.63** | **0.87** |

### 4.1.2. SEQUENCE LENGTH

**Experimental Settings** To evaluate the effectiveness of sequence length scaling for variance dampening, we train 1.2B models with varying maximum sequence lengths ranging from $T = 256$ to $T = 8192$ tokens. To isolate the effect of sequence length, we maintain constant computational budget across all configurations by adjusting the number of training steps inversely proportional to sequence length. All other hyperparameters remain identical across experiments. We analyze three key metrics: (1) the evolution of last-layer output variance throughout training, (2) the induced sparsity (defined in Section 3.2.1), and (3) validation perplexity and layer effectiveness scores. Detailed hyperparameters and training configurations are provided in Section C.3.

**Main Observation** Variance trajectories (Figure 5a) show consistent reduction with longer training sequences, with $T = 256$ exhibiting the highest variance. Measuring attention sparsity on held-out samples at $T = 8192$ (Figures 5b to 5d and 6), we observe increasing sparsity across all thresholds ($\epsilon \in \{10^{-3}, 10^{-4}, 10^{-6}\}$) as training length grows, confirming that longer contexts induce stronger implicit sparsity. Performance analysis (Table 3) reveals optimal scaling from $T = 256$ to $T = 2048$: perplexity improves

by 4+ points while usefulness increases (0.69→0.81). However, further scaling to $T = 8192$ yields diminishing returns despite lower variance, suggesting *over-dampening* that restricts model capacity similar to excessive weight decay (Li et al., 2025b; Izzo et al., 2025; Saada et al., 2024). Additional analysis of attention entropy and sparsity at different evaluation lengths is provided in Sections A.6 and A.7.

### 4.2. Explicit Sparsity

#### 4.2.1. GROUPED QUERY ATTENTION

**Experimental Settings** We analyze the variance dampening effect of GQA (Ainslie et al., 2023) by varying the number of key-value head groups $G$. We train 1.2B models with equal training FLOPs using group sizes $G \in \{1, 4, 16\}$ (MHA, GQA, and MQA (Shazeer, 2019), respectively). We evaluate the last-layer output variance, validation perplexity, and layer effectiveness scores. Detailed model definition and training configurations are provided in Section C.4.

**Main Observation** Results in Figure 7 show that variance decreases monotonically with group size. MQA ($G = 16$) exhibits 2× lower variance than MHA ($G = 1$). Under equal training FLOPs (Table 4), MQA achieves both better performance (PPL: 14.52 → 14.47) and higher layer effectiveness: usefulness score increases 7% (0.81 → 0.87), yielding 13 effective layers for MHA versus 14 for MQA.

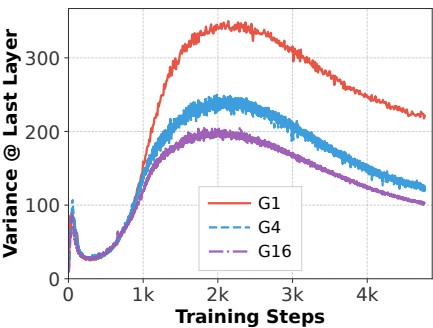

*Figure 7.* Output variance with different GQA configurations.

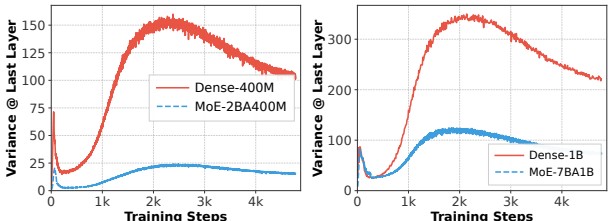

*(a)* 400M activated param.  *(b)* 1B activated param.

*Figure 8.* Comparison of last-layer variance between MoE and dense models with different activated parameters.

This confirms that key-value sharing reduces independent attention interactions and is associated with lower variance.

#### 4.2.2. MIXTURE OF EXPERTS

**Experimental Settings** We adopt fine-grained experts (Dai et al., 2024; Muennighoff et al., 2024) and evaluate two configurations: (1) 7B total parameters (1B active) with 64 experts and top-8 routing; (2) 2B total parameters (400M active) with 32 experts, top-4 routing, and one shared expert. We benchmark against dense baselines matched by active parameter count. Details are in Section C.5.

**Main Observation** We compare the variance of dense and MoE models in Figure 8. Results for both the 400M-active and 1B-active configurations show that the MoE architecture dampens variance by approximately 6× and 3×, respectively. Moreover, the MoE variants not only outperform their dense counterparts by over 2 perplexity but also exhibit higher layer effectiveness (e.g., usefulness score increases from 0.81 to 0.94 for the 1B configuration) (Table 5). These results demonstrate that explicit sparsity through MoE improves layer effectiveness.

## 5. Sparsity as an Enabler for Depth

Observing both implicit and explicit sparsity reduce variance, we explore scaling from $L = 16$ to $L = 32$ (at constant 1.2B parameters) while integrating different sparsity strategies to achieve better performance and layer utilization. All experiments use identical training data and parameter budgets, with learning rate sweeps for each configuration. We evaluate using average accuracy on ARC-C (Clark et al., 2018) and HellaSwag (Zellers et al., 2019), and Usefulness Score. Model and training details are in Section C.6. Figure 1 reveals the solution for effective depth scaling. Naively increasing depth from $L = 16$ to $L = 32$ degrades performance: accuracy drops by 0.5 while Usefulness Score plummets from 0.75 to 0.53, indicating approximately 50% layer underutilization. However, progressively integrating sparsity recovers and surpasses the baseline. Extending context to $T = 4096$ recovers accuracy (40.0) and improves utilization (0.59), though $T = 8192$ shows diminishing returns due to over-dampening. Introducing implicit sparsity

*Table 5.* Validation perplexity and comparison of layerwise effectiveness between dense and MoE models.

|  | PPL (↓) | Effectiveness Score | | |
|---|---|---|---|---|
|  |  | Causal | Permutation | Usefulness |
| Dense-400M | 17.56 | 0.30 | 0.54 | 0.87 |
| MoE-2BA400M | **15.89** | **0.36** | **0.59** | **0.94** |
| Dense-1B | 14.52 | 0.31 | 0.51 | 0.81 |
| MoE-7BA1B | **13.82** | **0.34** | **0.61** | **0.94** |

via weight decay also proves effective: $\lambda = 0.3$ increases accuracy to 41.4 with Usefulness Score rising to 0.63 (representing 20% improvement over the naive baseline). This trend continues with $\lambda = 0.6$ achieving competitive results, though excessive regularization ($\lambda = 1.0$) degrades both performance and utilization. Finally, combining explicit sparsity through GQA ($G = 2$) and MoE yields the best configuration: average accuracy of 44.1 and Usefulness Score of 0.75, representing a 4+ point accuracy gain over the $L = 16$ baseline while maintaining better layer utilization than the naive $L = 32$ model (0.75 vs. 0.53). These results demonstrate that appropriate sparsity mechanisms are essential for effective depth scaling.

## 6. Related Work

### 6.1. The Curse of Depth

While deeper networks achieve superior performance with sufficient data and compute (Kaplan et al., 2020; Hoffmann et al., 2022), recent studies reveal that modern Pre-LN LLMs (Yang et al., 2025; Bi et al., 2024; Dubey et al., 2024) suffer from increasing layer redundancy (Gromov et al., 2024; Men et al., 2025; Lad et al., 2024; Csordás et al., 2025). This degradation, termed the *Curse of Depth (CoD)* (Sun et al., 2025), stems from exponential variance growth that forces deeper layers toward identity mappings. Prior work addresses CoD through explicit variance control via scaled initialization (Takase et al., 2023; Zhang et al., 2019; Luther and Seung, 2019), normalization scaling (Sun et al., 2025; Zhu et al., 2025), or alternative normalization (Li et al., 2024; Cai et al., 2025). In contrast, we investigate whether sparsity from different choices can naturally mitigate CoD without explicit modifications.

## 6.2. Sparsity in Neural Networks

Sparsity has been central to neural network research for both biological inspiration (LeCun et al., 1989) and computational efficiency (Han et al., 2015). We categorize sparsity into two paradigms. **Implicit sparsity** emerges naturally from training dynamics and architectural choices: ReLU activations zero negative values (Glorot et al., 2011; Hayou et al., 2019), attention mechanisms concentrate on token subsets (Su et al., 2024; Xiao et al., 2023; Zhang et al., 2023; Yuan et al., 2025), and weight decay drives small parameters toward zero (Krogh and Hertz, 1991; Loshchilov and Hutter, 2019; Frankle and Carbin, 2018). **Explicit sparsity** is architecturally enforced to decouple capacity from computation: Mixture-of-Experts (MoE) activates only $k$ of $E$ experts per token (Fedus et al., 2022; Dai et al., 2024; Liu et al., 2025), while Grouped Query Attention (GQA) shares key-value projections across query heads (Ainslie et al., 2023; Shazeer, 2019; Dubey et al., 2024; Yang et al., 2025). While sparsity's efficiency benefits are well-established, its impact on variance propagation remains unexplored. We systematically provide evidence that these mechanisms are associated with reduced variance propagation and improved layer utilization in our LLM training settings.

## 7. Limitations

Our theoretical analysis omits several mechanisms central to contemporary Transformer training, including various normalization methods, correlations across attention heads, and the feedback between learned parameters and activation statistics. Consequently, our theorems are best viewed as qualitative statements about how interaction sparsity regulates variance growth, rather than as calibrated, end-to-end predictive models of the residual stream in fully trained LLMs.

Given this gap between our idealized analysis and deployed models, the empirical evidence presented in this paper serves to substantiate our broader claims. Across the training settings we evaluate, mechanisms that reduce effective interaction density or concentrate computation are consistently associated with lower residual-stream variance and improved layer effectiveness; we rely on these empirical regularities to bridge the gap between theory and practice.

## 8. Conclusion

We show that variance growth in deep networks leads to a *curse of depth*, where layers become increasingly underutilized as Jacobians drift toward identity mappings. We theoretically and empirically validate that sparsity, implicit or explicit, damps variance propagation, improving layer utilization and enabling effective depth scaling with better performance. This reframes sparsity as not only a compute-saving tool, but also an optimization mechanism that improves compute utilization by controling variance growth.

## Impact Statement

This paper presents methodological advances in understanding and mitigating the curse of depth in neural networks. We do not foresee direct negative societal impacts from this theoretical and empirical work on model architecture and training dynamics.

## Acknowledgement

This research was partially supported by the Deutsche Forschungsgemeinschaft (DFG) through the DFG Cluster of Excellence MATH+ (EXC-2046/1, EXC-2046/2, project id 390685689), as well as by the German Federal Ministry of Research, Technology and Space (research campus Modal, fund number 05M14ZAM, 05M20ZBM) and the VDI/VDE Innovation + Technik GmbH (fund number 16IS23025B).

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

# A. Additional Results

## A.1. Variance Across All Depths

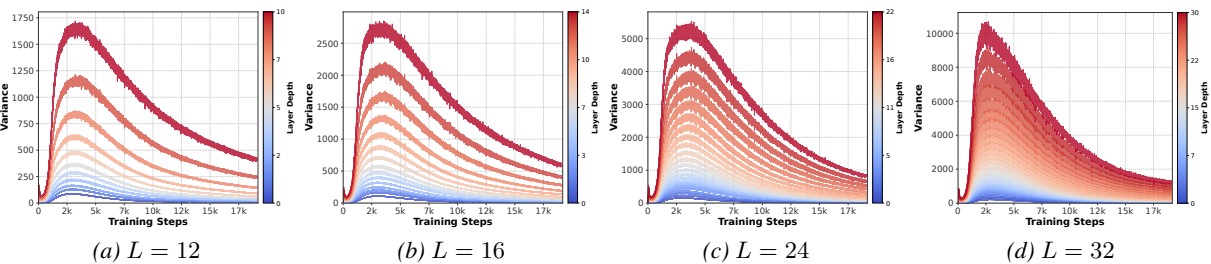

*(a)* $L = 12$      *(b)* $L = 16$      *(c)* $L = 24$      *(d)* $L = 32$

*Figure 9.* Variance of each layer's output during training for models of different depths in Section 2.

Instead of only the last layer's variance reported in previous settings, we also provide the variance of each layer during training for models with different depths in the depth control experiment in Section 2. The results are presented in Figure 9, where we can clearly see that the variance accumulates with depth.

## A.2. Variance of Different Blocks

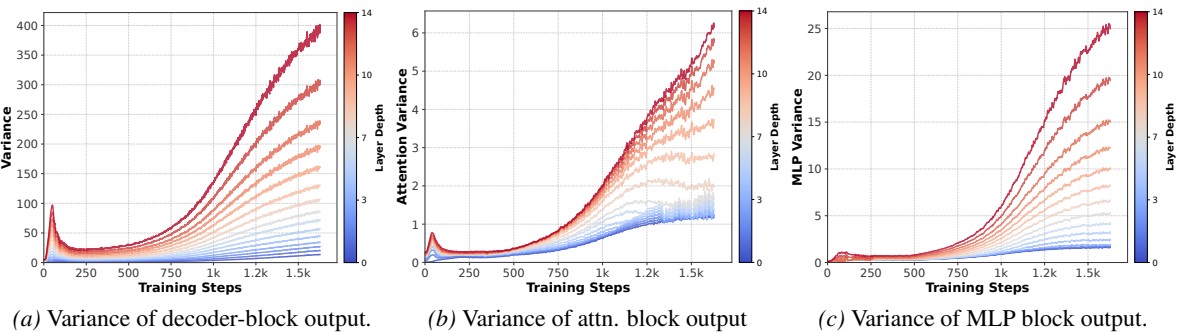

*(a)* Variance of decoder-block output.      *(b)* Variance of attn. block output      *(c)* Variance of MLP block output.

*Figure 10.* Variance of each layer's output during training with model $L = 16$.

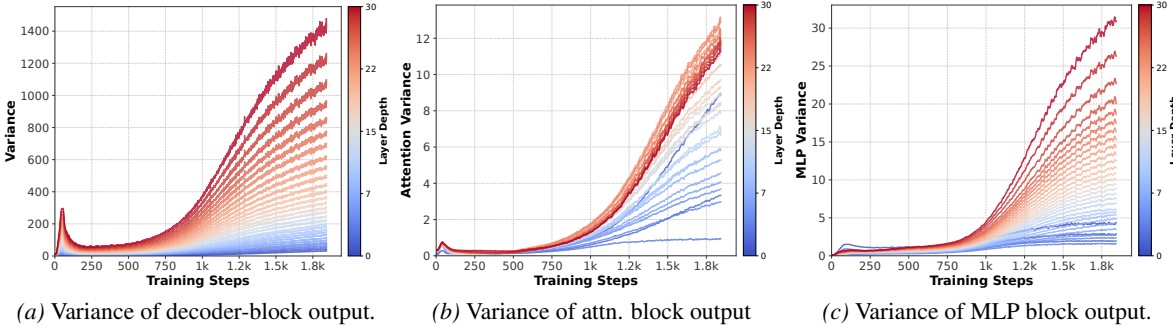

*(a)* Variance of decoder-block output.      *(b)* Variance of attn. block output      *(c)* Variance of MLP block output.

*Figure 11.* Variance of each layer's output during training with model $L = 32$.

Across all of our other experiments, we report the variance of each layer's output (e.g., the decoder layer in LLMs). In this subsection, we experiment with a model of $d = 1536$ hidden dimensions, MLP dimension of 4608, $h = 12$ attention heads, and maximum sequence length of 4096 with two depths: 16 and 32. We train for 1500+ steps using the FineWeb-Edu dataset with a learning rate of $1 \times 10^3$, weight decay of 0.1, and global batch size of 256. We report each layer's attention output variance and MLP output variance during training.

Results are presented in Figures 10 and 11. While output variance grows continuously with depth, MLP block variances increase more rapidly and show consistently stronger growth across depth compared to attention blocks. Interestingly, although the overall output variance exhibits substantial growth, the variance within the residual components (attention and

MLP blocks) themselves grows only moderately. This suggests that the shortcut paths may also contribute significantly to variance accumulation and may require regularization methods to control the variance growth, which we leave for future work.

### A.3. Jacobian Matrices of Different Layers

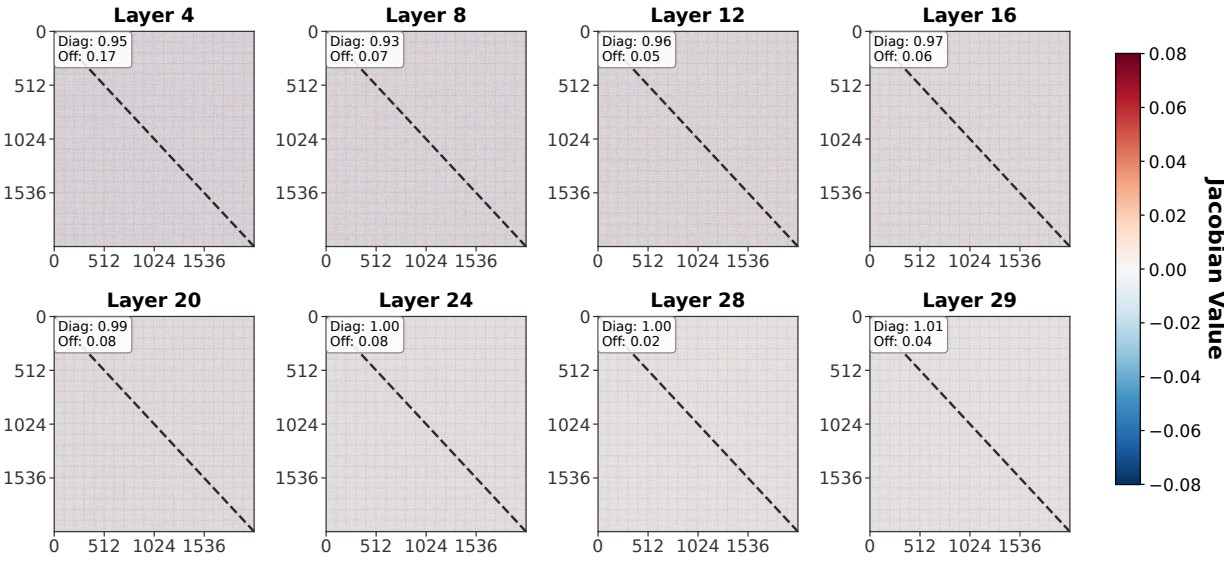

*Figure 12.* Jacobian matrices at different layers for $L = 32$. Results are transformed with a power of 0.3 for better visualization.

In Section 2, we prove that the Jacobian of the residual block approaches identity mapping with increased depth due to variance explosion. In Figure 3, we plot the Jacobian matrix at layer 30 for $L = 32$. Here, we plot Jacobian matrices at different layers for $L = 32$ in Figure 12, where we can see the off-diagonal components gradually decrease as depth increases.

### A.4. Kurtosis Analysis

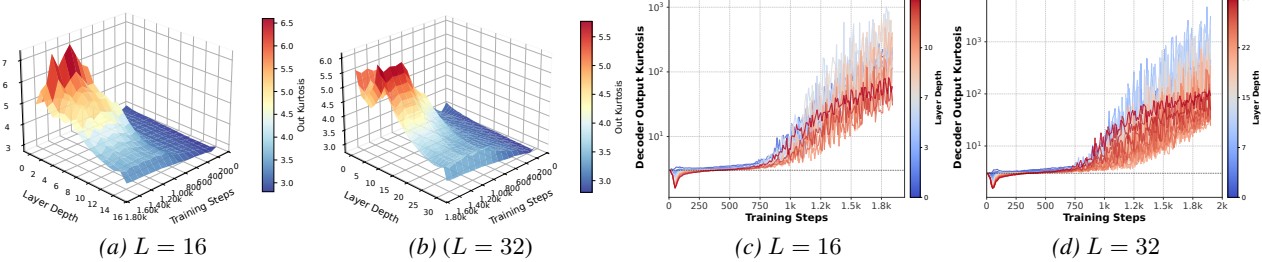

*Figure 13.* **(a)-(b):** per-dimension kurtosis. **(c)-(d):** layer-level kurtosis.

To investigate whether variance accumulation is driven by outlier features, we analyze kurtosis (fourth standardized moment) of layer outputs.

**Per-Dimension Kurtosis**   For each hidden dimension $j$ at layer $\ell$, we compute:

$$\text{Kurtosis}_{\ell,j} = \frac{\frac{1}{n}\sum_{i=1}^{n}(h_{\ell,i,j} - \bar{h}_{\ell,j})^4}{\left(\frac{1}{n}\sum_{i=1}^{n}(h_{\ell,i,j} - \bar{h}_{\ell,j})^2\right)^2} \tag{A.4.1}$$

where $\bar{h}_{\ell,j} = \frac{1}{n}\sum_{i=1}^{n} h_{\ell,i,j}$ is the mean across tokens for dimension $j$.

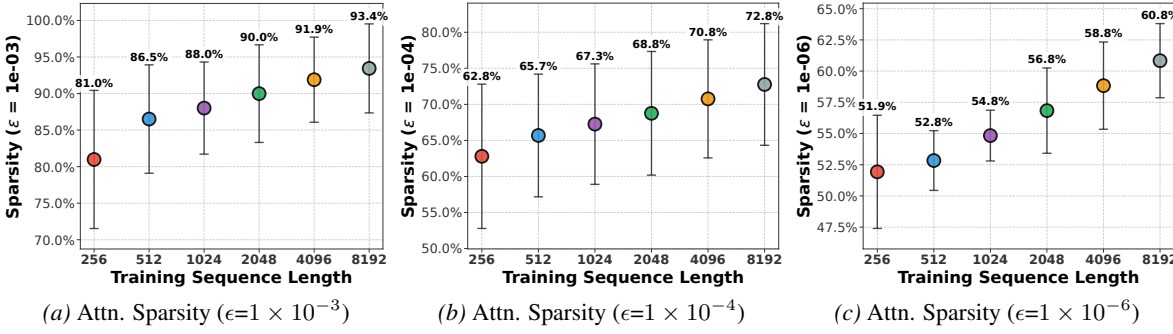

*(a)* Attn. Sparsity ($\epsilon$=1 × 10$^{-3}$)  *(b)* Attn. Sparsity ($\epsilon$=1 × 10$^{-4}$)  *(c)* Attn. Sparsity ($\epsilon$=1 × 10$^{-6}$)

*Figure 14.* Attention sparsity with evaluation length of 512.

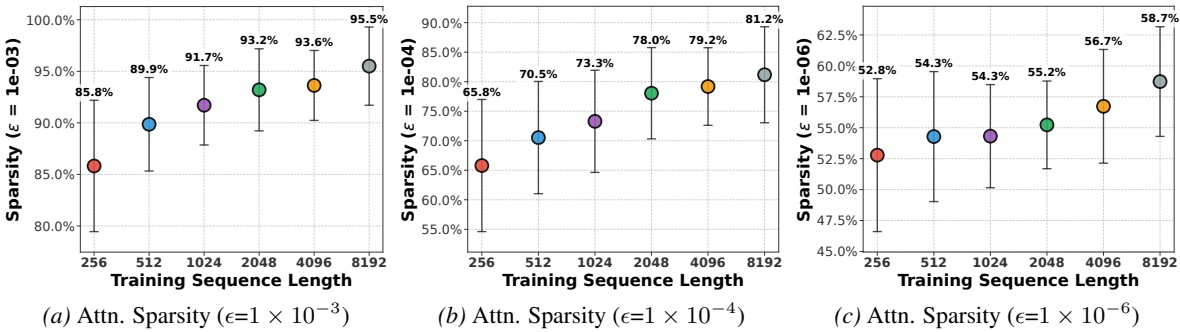

*(a)* Attn. Sparsity ($\epsilon$=1 × 10$^{-3}$)  *(b)* Attn. Sparsity ($\epsilon$=1 × 10$^{-4}$)  *(c)* Attn. Sparsity ($\epsilon$=1 × 10$^{-6}$)

*Figure 15.* Attention sparsity with evaluation length of 1024.

**Layer-Level Aggregation** We aggregate across dimensions to obtain layer-level kurtosis:

$$\text{Kurtosis}_\ell = \frac{1}{d} \sum_{j=1}^{d} \text{Kurtosis}_{\ell,j} \tag{A.4.2}$$

We compute both per-dimension kurtosis and layer-level kurtosis for all layers in the configurations from Section A.2 during training. The results are presented in Figure 13. From both the layer-level and per-dimension kurtosis across different depths, we can see that early layers show more outlier features (higher kurtosis) compared to deeper layers, which demonstrates that variance explosion is not driven by outlier features. We hypothesize that since prior work has shown that outlier features are crucial for model performance (Lin et al., 2024), the lower outlier prevalence in deeper layers indicates that they learn less effectively compared to shallower layers.

## A.5. Attention Sparsity Across Different Evaluation Lengths

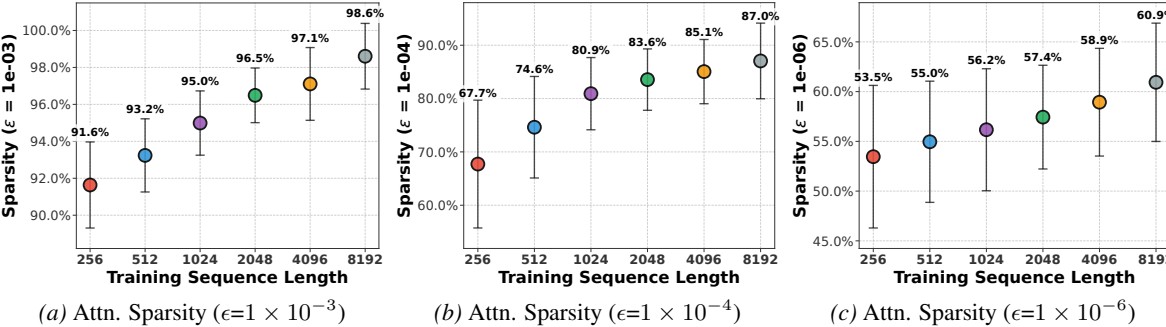

*(a)* Attn. Sparsity ($\epsilon$=1 × 10$^{-3}$)  *(b)* Attn. Sparsity ($\epsilon$=1 × 10$^{-4}$)  *(c)* Attn. Sparsity ($\epsilon$=1 × 10$^{-6}$)

*Figure 16.* Attention sparsity with evaluation length of 2048.

In Figure 5 and Figure 6, we evaluate attention sparsity at a fixed evaluation length of 8192 tokens across all models trained with different sequence lengths. To further investigate how sparsity varies when evaluation length matches training length,

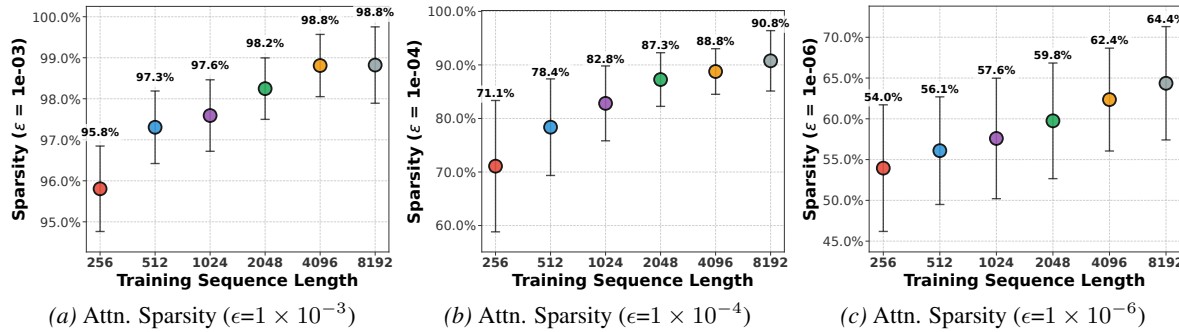

*Figure 17.* Attention sparsity with evaluation length of 4096.

we provide additional results at evaluation lengths of 512, 1024, 2048, and 4096 tokens (Figures 14 to 17). Across all evaluation lengths, sparsity increases consistently with training sequence length, validating that longer training sequences induce stronger implicit attention sparsity regardless of evaluation context.

## A.6. Attention Entropy

Complementing threshold-based sparsity (Equation (3.6)), we measure attention concentration using entropy. For attention weights $\mathbf{A}_{\ell,h} \in \mathbb{R}^{T \times T}$ at layer $\ell$ and head $h$, per-query entropy is:

$$\text{Entropy}_{\ell,h}(i) = -\sum_{j=1}^{T} A_{i,j} \log A_{i,j}. \tag{A.6.1}$$

Head-level entropy averages across queries:

$$\text{Entropy}_{\ell,h} = \frac{1}{T} \sum_{i=1}^{T} \text{Entropy}_{\ell,h}(i). \tag{A.6.2}$$

Global entropy averages across all layers ($L$) and heads ($H$):

$$\text{Entropy}_{\text{global}} = \frac{1}{L \cdot H} \sum_{\ell=1}^{L} \sum_{h=1}^{H} \text{Entropy}_{\ell,h} \tag{A.6.3}$$

Lower entropy indicates concentrated attention (sparse).

Attention entropy (Figure 18) decreases with training length across all evaluation length, mirroring the sparsity results and confirming that longer training induces stronger implicit sparsity.

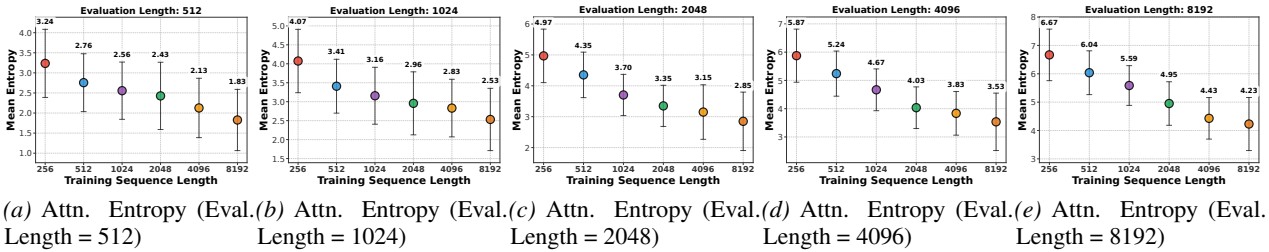

*(a)* Attn. Entropy (Eval. Length = 512)   *(b)* Attn. Entropy (Eval. Length = 1024)   *(c)* Attn. Entropy (Eval. Length = 2048)   *(d)* Attn. Entropy (Eval. Length = 4096)   *(e)* Attn. Entropy (Eval. Length = 8192)

*Figure 18.* Attention entropy with different evaluation length.

## A.7. Extended Variance Trajectories

Figures 5a and 7 show last-layer variance trajectories during training for models with different sequence lengths and group sizes. Since we control for total training FLOPs, models with shorter sequences require more training steps to match the compute of longer-sequence models, and models with larger group sizes (fewer KV heads) similarly require more steps to match FLOPs. For visualization clarity, we align trajectories to the run with fewest training steps in Figure 5a; complete unaligned trajectories are provided in Figure 19.

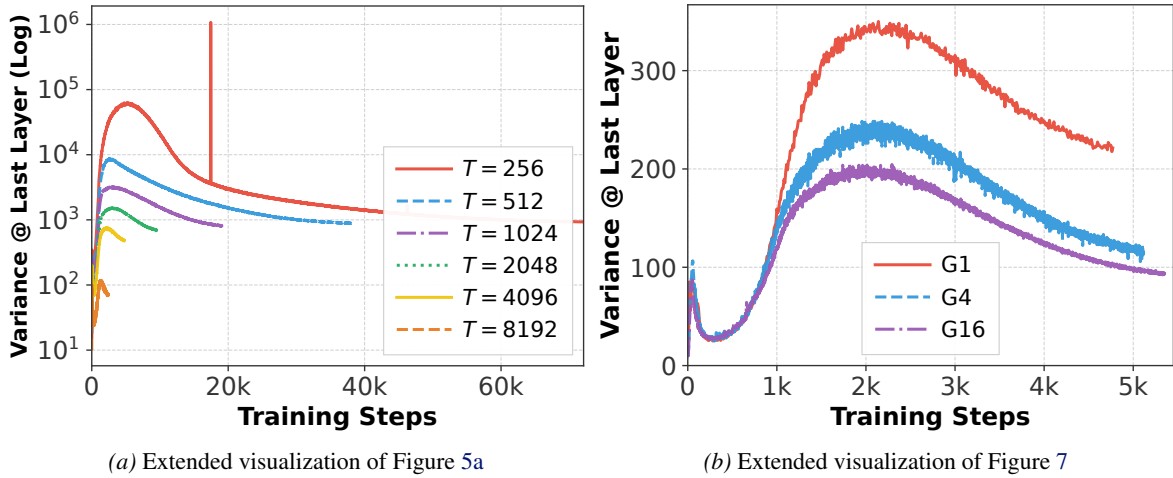

*(a)* Extended visualization of Figure 5a        *(b)* Extended visualization of Figure 7

*Figure 19.* Variance dynamics throughout training for models with different sequence lengths and group sizes. All models trained with equal FLOPs within its comparison group, leading to different training durations.

## A.8. Sensitivity Analysis

We compute the usefulness score (Equation (2.10)) using a default threshold of $\alpha = 0.1$. In this section, we analyze the sensitivity of this metric to varying threshold values. Specifically, we train models at depths of $L = 16$ and $L = 32$, applying sequence lengths of $T = 1024$ and $T = 4096$ to each, and evaluate the resulting usefulness scores under these different configurations.

The results in Table 6 demonstrate that the relative ordering and broader trends remain highly consistent. Deeper models consistently exhibit a greater number of low-effectiveness layers regardless of the chosen $\alpha$, while the introduction of sparsity improves layer usefulness in these configurations. Furthermore, our choice of $\alpha = 0.1$ is empirically justified by the natural bimodal distribution of per-layer loss increases observed throughout our experiments. Layers tend to strictly bifurcate into two categories: clearly useful (yielding loss increases far exceeding 10%) or clearly redundant (yielding loss increases well below 10%), leaving very few layers near the decision boundary.

*Table 6.* Sensitivity analysis for suefullness threshold in Equation (2.10).

| Depth | Sequence Length | Usefull. @ $\alpha = 0.05$ | Usefull. @ $\alpha = 0.1$ | Usefull. @ $\alpha = 0.2$ |
|---|---|---|---|---|
| $L = 16$ | 1024 | 0.88 | 0.75 | 0.63 |
| $L = 16$ | 4096 | **1.00** | **0.81** | **0.75** |
| $L = 32$ | 1024 | 0.72 | 0.56 | 0.44 |
| $L = 32$ | 4096 | **0.75** | **0.59** | **0.56** |

## A.9. Model Initialization

Throughout our experiments, we default to training our models using a standard normal initialization ($\mu = 0, \sigma = 0.02$) for all linear layers. In this section, we investigate the effects of adopting the scaled initialization proposed by (Zhang et al., 2019). To this end, we train a model of depth $L = 16$ under two different initialization schemes: standard normal and scaled initialization.

The results presented in Table 7 demonstrate that while scaled initialization effectively dampens variance propagation, the introduction of sparsity (achieved here via larger sequence lengths) provides an additional benefit, further enhancing depth effectiveness and leading to superior overall performance.

*Table 7.* Analysis of different initialization method.

| Depth | Sequence Length | Initialization | Var. @ Last Layer | Usefulness | Loss |
|---|---|---|---|---|---|
| $L = 16$ | 1024 | Normal | $\approx$3e3 | 0.81 | 2.70 |
| $L = 16$ | 1024 | Scaled | $\approx$**2e3** | 0.81 | 2.71 |
| $L = 16$ | 4096 | Scaled | $\approx$**5e2** | **1.00** | **2.64** |

# B. Proofs and Mathematical background

## B.1. Proof of Theorem 1

*Proof.* For $\ell \in \{0, \dots, L-1\}$ and $r_{\ell+1} = r_\ell + u_\ell$ with $u_\ell = W_\ell(D_\ell r_\ell)$. Assume $\mathbb{E}[r_\ell] = 0$, because $W_\ell$ is independent of $(r_\ell, D_\ell)$, we have:

$$\text{Var}(r_{\ell+1}) = \text{Var}(r_\ell) + \text{Var}(u_\ell) + 2\mathbb{E}\langle r_\ell, u_\ell \rangle. \tag{B.1.1}$$

By definition $u_\ell = W_\ell(D_\ell r_\ell)$. Condition on $(r_\ell, D_\ell)$ and apply (3.2) with $x = D_\ell r_\ell$:

$$\mathbb{E}\left[\|u_\ell\|_2^2 \mid r_\ell, D_\ell\right] = \mathbb{E}\left[\|W_\ell(D_\ell r_\ell)\|_2^2 \mid r_\ell, D_\ell\right] \leq \alpha_\ell \|D_\ell r_\ell\|_2^2. \tag{B.1.2}$$

Take expectation over $(r_\ell, D_\ell)$:

$$\text{Var}(u_\ell) = \mathbb{E}\|u_\ell\|_2^2 \leq \alpha_\ell \mathbb{E}\|D_\ell r_\ell\|_2^2. \tag{B.1.3}$$

Next, condition on $r_\ell$ and apply (3.2) to $x = r_\ell$:

$$\mathbb{E}\left[\|D_\ell r_\ell\|_2^2 \mid r_\ell\right] \leq \rho_\ell \|r_\ell\|_2^2. \tag{B.1.4}$$

Take expectation over $r_\ell$ and combine with (B.1.3):

$$\text{Var}(u_\ell) \leq \alpha_\ell \rho_\ell \mathbb{E}\|r_\ell\|_2^2 = \alpha_\ell \rho_\ell \text{Var}(r_\ell). \tag{B.1.5}$$

By Cauchy-Schwarz inequality,

$$\mathbb{E}\langle r_\ell, u_\ell \rangle \leq \mathbb{E}\left[\|r_\ell\|_2\|u_\ell\|_2\right] \leq \sqrt{\mathbb{E}\|r_\ell\|_2^2}\sqrt{\mathbb{E}\|u_\ell\|_2^2}$$
$$= \sqrt{\text{Var}(r_\ell)}\sqrt{\text{Var}(u_\ell)}. \tag{B.1.6}$$

Using (B.1.5), hence

$$\mathbb{E}\langle r_\ell, u_\ell \rangle \leq \sqrt{\alpha_\ell \rho_\ell}\text{Var}(r_\ell). \tag{B.1.7}$$

Plug (B.1.5) and (B.1.7) into (B.1.1):

$$\text{Var}(r_{\ell+1}) \leq \text{Var}(r_\ell) + \alpha_\ell \rho_\ell \text{Var}(r_\ell) + 2\sqrt{\alpha_\ell \rho_\ell}\text{Var}(r_\ell)$$
$$= \left(1 + \sqrt{\alpha_\ell \rho_\ell}\right)^2 \text{Var}(r_\ell). \tag{B.1.8}$$

Iterate (B.1.8) for $\ell = 0, \dots, L-1$:

$$\text{Var}(r_L) \leq \text{Var}(r_0) \prod_{\ell=0}^{L-1} \left(1 + \sqrt{\alpha_\ell \rho_\ell}\right)^2 \tag{B.1.9}$$

In particular, if $\alpha_\ell \leq \alpha$ and $\rho_\ell \leq \rho < 1$ for all $\ell$, then

$$\text{Var}(r_L) \leq \text{Var}(r_0)\left(1 + \sqrt{\alpha\rho}\right)^{2L}$$
$$= O\left(1 + \sqrt{\alpha\rho}\right)^{2L} \tag{B.1.10}$$

so smaller $\rho$ (more sparsity) gives a smaller per-layer variance gain. which is (3.3). The uniform bound with $(\alpha, \rho)$ follows directly. $\square$

Theorem 1 depends on sparsity only through $\rho_\ell$, the (second-moment) energy retention of the mask $D_\ell$. Any mechanism that makes the effective mask sparser (smaller $\rho_\ell$), even if it is not an explicit architectural constraint, yields a smaller per-layer gain $\left(1 + \sqrt{\alpha_\ell \rho_\ell}\right)^2$ and thus a smaller global variance growth across depth.

## B.2. Attention Based Sparsity as variance Regularizer

Let $R_\ell \in \mathbb{R}^{n \times d}$ be the token matrix at 2 $\ell$. Fix a *single* feature mask $D_\ell \in \mathbb{R}^{d \times d}$ (diagonal, 0–1), which is used by both the attention and FFN sublayers at depth $\ell$. Define the two residual sublayers (no LayerNorm) in the standard sequential form

$$Z_\ell = R_\ell + \text{MHA}_\ell(R_\ell D_\ell), \qquad R_{\ell+1} = Z_\ell + \text{FFN}_\ell(Z_\ell D_\ell), \tag{B.2.1}$$

for $\ell = 0, \ldots, L-1$.

Let the number of heads be $H$, head width be $d_h$ with $H d_h = d$. For $X \in \mathbb{R}^{n \times d}$, define for each head $h \in \{1, \ldots, H\}$:

$$Q_{\ell,h}(X) = X W_{\ell,h}^Q, \quad K_{\ell,h}(X) = X W_{\ell,h}^K, \quad V_{\ell,h}(X) = X W_{\ell,h}^V, \tag{B.2.2}$$

with $W_{\ell,h}^Q, W_{\ell,h}^K, W_{\ell,h}^V \in \mathbb{R}^{d \times d_h}$. Let

$$S_{\ell,h}(X) = \frac{Q_{\ell,h}(X) K_{\ell,h}(X)^\top}{\sqrt{d_h}} \in \mathbb{R}^{n \times n}, \qquad P_{\ell,h}(X) = \text{Softmax}\big(S_{\ell,h}(X)\big) \in \mathbb{R}^{n \times n}, \tag{B.2.3}$$

where Softmax applies softmax to each row. The head output is

$$H_{\ell,h}(X) = P_{\ell,h}(X) V_{\ell,h}(X) \in \mathbb{R}^{n \times d_h}. \tag{B.2.4}$$

Concatenate heads along features:

$$H_\ell(X) = \big[H_{\ell,1}(X); \ldots; H_{\ell,H}(X)\big] \in \mathbb{R}^{n \times (H d_h)} = \mathbb{R}^{n \times d}, \tag{B.2.5}$$

and apply the output projection $W_\ell^O \in \mathbb{R}^{d \times d}$:

$$\text{MHA}_\ell(X) = H_\ell(X) W_\ell^O \in \mathbb{R}^{n \times d}. \tag{B.2.6}$$

Theorem 2 establishes that, for a residual recursion, variance propagation across depth is controlled by a per-layer gain factor that depends on sparsity only through a second-moment **energy retention** parameter. We now specialize this principle to a Transformer residual block composed of *multi-head self-attention* (MHA) and *feed-forward network* (FFN), *without LayerNorm*. Unlike the linear ResNet recursion, MSA includes the nonlinear Softmax coupling across tokens. Our goal is to isolate a sufficient condition under which the same conclusion holds: using a common feature mask $D_\ell$ for both MSA and FFN yields a variance bound whose dependence on sparsity appears only via the mask retention factor $\rho_\ell$.

**Assumption 1** (Common sparsity mask with bounded energy retention). *For each layer $\ell$, there exists a diagonal 0–1 mask $D_\ell \in \mathbb{R}^{d \times d}$ shared by the MSA and FFN sublayers, and a constant $\rho_\ell \in [0,1]$ such that for all $X \in \mathbb{R}^{n \times d}$,*

$$\mathbb{E}\|X D_\ell\|_F^2 \le \rho_\ell \|X\|_F^2. \tag{B.2.7}$$

**Assumption 2** (MSA second-moment gain via explicit $QKV/O$ control). *For each layer $\ell$ and head $h \in \{1, \ldots, H\}$, let*

$$Q_{\ell,h}(X) = X W_{\ell,h}^Q, \; K_{\ell,h}(X) = X W_{\ell,h}^K, \; V_{\ell,h}(X) = X W_{\ell,h}^V, \; P_{\ell,h}(X) = \text{Softmax}\left(\frac{Q_{\ell,h}(X) K_{\ell,h}(X)^\top}{\sqrt{d_h}}\right), \tag{B.2.8}$$

*and define* $\text{MHA}_\ell(X) = \big[P_{\ell,1}(X) V_{\ell,1}(X); \ldots; P_{\ell,H}(X) V_{\ell,H}(X)\big] W_\ell^O$. *Assume there exist nonnegative constants $\kappa_{\ell,h}$, $\nu_{\ell,h}$, and $\omega_\ell$ such that for all $X \in \mathbb{R}^{n \times d}$,*

$$\|P_{\ell,h}(X)\|_2 \le \kappa_{\ell,h}, \qquad \|W_{\ell,h}^V\|_2 \le \nu_{\ell,h}, \qquad \|W_\ell^O\|_2 \le \omega_\ell. \tag{B.2.9}$$

*Define*

$$\alpha_\ell^{\text{attn}} := \omega_\ell^2 \sum_{h=1}^H \kappa_{\ell,h}^2 \nu_{\ell,h}^2. \tag{B.2.10}$$

**Assumption 3** (FFN second-moment gain). *For each layer $\ell$, $\text{FFN}_\ell : \mathbb{R}^{n \times d} \to \mathbb{R}^{n \times d}$ satisfies: there exists $\alpha_\ell^{\text{ffn}} \ge 0$ such that for all $X \in \mathbb{R}^{n \times d}$,*

$$\|\text{FFN}_\ell(X)\|_F^2 \le \alpha_\ell^{\text{ffn}} \|X\|_F^2. \tag{B.2.11}$$

*For example, if $\text{FFN}_\ell(X) = \sigma(X W_{\ell,1}) W_{\ell,2}$ with $\sigma$ being $L_\sigma$-Lipschitz and $\|W_{\ell,1}\|_2 \le s_{\ell,1}$, $\|W_{\ell,2}\|_2 \le s_{\ell,2}$, then one may take $\alpha_\ell^{\text{ffn}} = (L_\sigma s_{\ell,1} s_{\ell,2})^2$.*

**Theorem 2** (Sparsity reduces variance propagation in Transformer). *Let $\{R_\ell\}_{\ell=0}^L$ follow the no-LayerNorm Transformer recursion* (B.2.1). *Under Assumptions 1–3, the depth-$L$ variance satisfies*

$$\mathrm{Var}(R_L) \le \mathrm{Var}(R_0) \prod_{\ell=0}^{L-1} \left(1 + \sqrt{\alpha_\ell^{\mathrm{attn}}\rho_\ell}\right)^2 \left(1 + \sqrt{\alpha_\ell^{\mathrm{ffn}}\rho_\ell}\right)^2. \tag{B.2.12}$$

*In particular, if $\alpha_\ell^{\mathrm{attn}} \le \alpha^{\mathrm{attn}}$, $\alpha_\ell^{\mathrm{ffn}} \le \alpha^{\mathrm{ffn}}$ and $\rho_\ell \le \rho < 1$ for all $\ell$, then*

$$\begin{aligned}
\mathrm{Var}(R_L) &\le \mathrm{Var}(R_0)\left(1 + \sqrt{\alpha^{\mathrm{attn}}\rho}\right)^{2L}\left(1 + \sqrt{\alpha^{\mathrm{ffn}}\rho}\right)^{2L} \\
&= O\left(\left(1 + \sqrt{\alpha^{\mathrm{ffn}}\rho}\right)^{2L}\left(1 + \sqrt{\alpha^{\mathrm{attn}}\rho}\right)^{2L}\right).
\end{aligned} \tag{B.2.13}$$

The proof is provided in Section B.3. Theorem 2 shows that, even with the nonlinear Softmax in MHA, the depth-wise variance propagation can be controlled by a multiplicative gain whose dependence on sparsity appears only through the common retention factor $\rho_\ell$. In this sense, feature sparsity acts as a variance regularizer: smaller $\rho_\ell$ reduces both the attention-branch gain $(1 + \sqrt{\alpha_\ell^{\mathrm{attn}}\rho_\ell})^2$ and the FFN-branch gain $(1 + \sqrt{\alpha_\ell^{\mathrm{ffn}}\rho_\ell})^2$, hence decreases the overall variance growth across depth. The theorem therefore applies directly only to mechanisms that can be represented by such an independent mask model. For training-induced or data-dependent sparsity, the same expression should be viewed as an analogy based on an empirical retention factor, not as a formal implication. When effective sparsity is induced by training dynamics, this interpretation becomes heuristic because the mask, activations, and weights are coupled.

## B.3. Proof of Theorem 2

*Proof.* Let $U_\ell^{\mathrm{attn}} := \mathrm{MHA}_\ell(R_\ell D_\ell)$, so $Z_\ell = R_\ell + U_\ell^{\mathrm{attn}}$. Expand

$$\mathrm{Var}(Z_\ell) = \mathbb{E}\|Z_\ell\|_F^2 = \mathbb{E}\|R_\ell\|_F^2 + \mathbb{E}\|U_\ell^{\mathrm{attn}}\|_F^2 + 2\mathbb{E}\langle R_\ell, U_\ell^{\mathrm{attn}}\rangle_F, \tag{B.3.1}$$

where $\langle A, B\rangle_F = \mathrm{trace}(A^\top B)$.

We first bound $\mathbb{E}\|U_\ell^{\mathrm{attn}}\|_F^2$ using the explicit $QKV/O$ structure. Fix any $X \in \mathbb{R}^{n \times d}$ and consider $\mathrm{MHA}_\ell(X)$. By (B.2.6) and the spectral norm bound on $W_\ell^O$,

$$\begin{aligned}
\|\mathrm{MHA}_\ell(X)\|_F &= \|H_\ell(X)W_\ell^O\|_F \le \|W_\ell^O\|_2\|H_\ell(X)\|_F \\
&\le \omega_\ell\|H_\ell(X)\|_F.
\end{aligned} \tag{B.3.2}$$

By (B.2.5), $\|H_\ell(X)\|_F^2 = \sum_{h=1}^H \|H_{\ell,h}(X)\|_F^2$. For each head, using (B.2.4) and $\|AB\|_F \le \|A\|_2\|B\|_F$,

$$\begin{aligned}
\|H_{\ell,h}(X)\|_F &= \|P_{\ell,h}(X)V_{\ell,h}(X)\|_F \le \|P_{\ell,h}(X)\|_2\|V_{\ell,h}(X)\|_F \\
&\le \kappa_{\ell,h}\|V_{\ell,h}(X)\|_F.
\end{aligned} \tag{B.3.3}$$

Also, $V_{\ell,h}(X) = XW_{\ell,h}^V$ and $\|XW\|_F \le \|W\|_2\|X\|_F$, so

$$\|V_{\ell,h}(X)\|_F \le \|W_{\ell,h}^V\|_2\|X\|_F \le \nu_{\ell,h}\|X\|_F. \tag{B.3.4}$$

Combining (B.3.2)–(B.3.4) yields

$$\begin{aligned}
\|\mathrm{MHA}_\ell(X)\|_F^2 &\le \omega_\ell^2 \sum_{h=1}^H \kappa_{\ell,h}^2 \nu_{\ell,h}^2 \|X\|_F^2 \\
&= \alpha_\ell^{\mathrm{attn}}\|X\|_F^2,
\end{aligned} \tag{B.3.5}$$

with $\alpha_\ell^{\mathrm{attn}}$ from (B.2.10). Applying (B.3.5) to $X = R_\ell D_\ell$ and taking expectation gives

$$\mathbb{E}\|U_\ell^{\mathrm{attn}}\|_F^2 = \mathbb{E}\|\mathrm{MHA}_\ell(R_\ell D_\ell)\|_F^2 \le \alpha_\ell^{\mathrm{attn}}\mathbb{E}\|R_\ell D_\ell\|_F^2. \tag{B.3.6}$$

Using the Assumption 1 with $X = R_\ell$ yields

$$\mathbb{E}\|R_\ell D_\ell\|_F^2 \le \rho_\ell\mathbb{E}\|R_\ell\|_F^2 = \rho_\ell\mathrm{Var}(R_\ell), \tag{B.3.7}$$

so

$$\mathbb{E}\|U_\ell^{\mathrm{attn}}\|_F^2 \leq \alpha_\ell^{\mathrm{attn}} \rho_\ell \mathrm{Var}(R_\ell). \tag{B.3.8}$$

Using Cauchy–Schwarz gives

$$\begin{aligned} \mathbb{E}\langle R_\ell, U_\ell^{\mathrm{attn}}\rangle_F &\leq \sqrt{\mathbb{E}\|R_\ell\|_F^2} \sqrt{\mathbb{E}\|U_\ell^{\mathrm{attn}}\|_F^2} \\ &\leq \sqrt{\alpha_\ell^{\mathrm{attn}} \rho_\ell} \mathrm{Var}(R_\ell). \end{aligned} \tag{B.3.9}$$

Substitute (B.3.8) and (B.3.9) into (B.3.1) to obtain

$$\mathrm{Var}(Z_\ell) \leq \left(1 + \sqrt{\alpha_\ell^{\mathrm{attn}} \rho_\ell}\right)^2 \mathrm{Var}(R_\ell). \tag{B.3.10}$$

Following Theorem 1, let $U_\ell^{\mathrm{ffn}} := \mathrm{FFN}_\ell(Z_\ell D_\ell)$. Repeating the same variance expansion and Cauchy–Schwarz argument, and using (3) plus (1), we get

$$\mathrm{Var}(R_{\ell+1}) \leq \left(1 + \sqrt{\alpha_\ell^{\mathrm{ffn}} \rho_\ell}\right)^2 \mathrm{Var}(Z_\ell). \tag{B.3.11}$$

Combine (B.3.10) and (B.3.11):

$$\begin{aligned} \mathrm{Var}(R_{\ell+1}) &\leq \left(1 + \sqrt{\alpha_\ell^{\mathrm{attn}} \rho_\ell}\right)^2 \left(1 + \sqrt{\alpha_\ell^{\mathrm{ffn}} \rho_\ell}\right)^2 \mathrm{Var}(R_\ell) \\ &= O\left(\left(1 + \sqrt{\alpha^{\mathrm{ffn}} \rho}\right)^{2L} \left(1 + \sqrt{\alpha^{\mathrm{attn}} \rho}\right)^{2L}\right). \end{aligned} \tag{B.3.12}$$

Iterating (B.3.12) for $\ell = 0, \ldots, L-1$ yields (B.2.12). The uniform bound (B.2.13) follows by $\alpha_\ell^{\mathrm{attn}} \leq \alpha^{\mathrm{attn}}$, $\alpha_\ell^{\mathrm{ffn}} \leq \alpha^{\mathrm{ffn}}$, and $\rho_\ell \leq \rho$. □

## B.4. Weight Decay Variance Control: Theory and Proof

We provide the formal theoretical analysis showing that weight decay contracts parameter variance and reduces layer output variance. This result is a linearized variance-control calculation. In full Transformer training, gradients, weights, and activations are coupled, so the theorem should be interpreted as explaining one possible regularizing effect of weight decay rather than as a complete model.

**Theorem 3** (Weight decay contracts parameter variance and reduces layer-output variance). *Let $W_t \in \mathbb{R}^{d_{\mathrm{out}} \times d_{\mathrm{in}}}$ follow the decoupled weight-decay update $W_{t+1} = (1 - \eta\lambda)W_t - \eta G_t$, Assume $0 < \eta\lambda < 2$, and that the gradient noise is independent of the weight. If $\mathrm{Var}(G_t) \leq \sigma_G^2$ for all t, then for a linear layer output $u_t$, we have*

$$\begin{aligned} \mathrm{Var}(u_t) &\leq \|\Sigma_x\|_2 \left(\mathrm{Var}(W_t) + \|\mathbb{E}[W_t]\|_F^2\right), \\ &= O\left((1 - \eta\lambda)^{2t} + \frac{\eta\sigma_G^2}{\lambda}\right). \end{aligned} \tag{B.4.1}$$

*where $\Sigma_x = \mathbb{E}[xx^\top]$ is the covariance matrix of x, and x independent of $W_t$*

Under $0 < \eta\lambda \leq 1$, the bound in (B.4.1) is decreasing in $\lambda$ term-by-term: $(1 - \eta\lambda)^{2t}$ decreases monotonically with $\lambda$ and captures contraction of the initialization contribution, while $\eta\sigma_G^2/\lambda$ decreases with $\lambda$ and captures the steady-state variance induced by gradient noise. Hence, larger weight decay $\lambda$ yields a smaller upper bound on $\mathrm{Var}(u_t)$. Since weight decay is applied during training and continuously shrinks parameter magnitude, this variance reduction can be viewed as an implicit regularization effect induced by optimization.

*Proof.* Let $\bar{W}_t := \mathbb{E}[W_t]$ and $\widetilde{W}_t := W_t - \bar{W}_t$, and similarly $\bar{G}_t := \mathbb{E}[G_t]$, $\widetilde{G}_t := G_t - \bar{G}_t$. Taking expectation of $W_{t+1} = (1 - \eta\lambda)W_t - \eta G_t$ gives

$$\bar{W}_{t+1} = (1 - \eta\lambda)\bar{W}_t - \eta\bar{G}_t. \tag{B.4.2}$$

Subtract (B.4.2) from $W_{t+1} = (1 - \eta\lambda)W_t - \eta G_t$ to obtain the centered update

$$\widetilde{W}_{t+1} = (1 - \eta\lambda)\widetilde{W}_t - \eta\widetilde{G}_t. \tag{B.4.3}$$

By definition,

$$\text{Var}(W_{t+1}) = \mathbb{E}\left[\|\widetilde{W}_{t+1}\|_F^2\right] = \mathbb{E}\left[\|(1-\eta\lambda)\widetilde{W}_t - \eta\widetilde{G}_t\|_F^2\right]. \tag{B.4.4}$$

Thus we have:

$$\|(1-\eta\lambda)\widetilde{W}_t - \eta\widetilde{G}_t\|_F^2 = (1-\eta\lambda)^2\|\widetilde{W}_t\|_F^2 + \eta^2\|\widetilde{G}_t\|_F^2 - 2\eta(1-\eta\lambda)\langle\widetilde{W}_t, \widetilde{G}_t\rangle_F. \tag{B.4.5}$$

Taking expectation and by the assumption that $W_t$ and $G_t$ are independent:

$$\begin{aligned} \text{Var}(W_{t+1}) &= (1-\eta\lambda)^2\mathbb{E}\left[\|\widetilde{W}_t\|_F^2\right] + \eta^2\mathbb{E}\left[\|\widetilde{G}_t\|_F^2\right] \\ &= (1-\eta\lambda)^2\text{Var}(W_t) + \eta^2\text{Var}(G_t), \end{aligned} \tag{B.4.6}$$

Let $\rho := (1-\eta\lambda)^2$. Under $0 < \eta\lambda < 2$, we have $0 \le \rho < 1$. If $\text{Var}(G_t) \le \sigma_G^2$, then we have:

$$\text{Var}(W_{t+1}) \le \rho\text{Var}(W_t) + \eta^2\sigma_G^2. \tag{B.4.7}$$

Iterating the inequality gives

$$\begin{aligned} \text{Var}(W_t) &\le \rho^t\text{Var}(W_0) + \eta^2\sigma_G^2\sum_{s=0}^{t-1}\rho^s \\ &= \rho^t\text{Var}(W_0) + \eta^2\sigma_G^2\frac{1-\rho^t}{1-\rho} \\ &\le \rho^t\text{Var}(W_0) + \frac{\eta^2\sigma_G^2}{1-\rho}, \end{aligned} \tag{B.4.8}$$

Let $u_t = W_t x$. Since $\mathbb{E}[x] = 0$ and $x$ is independent of $W_t$,

$$\begin{aligned} \mathbb{E}[u_t] &= \mathbb{E}[W_t]\mathbb{E}[x] = 0, \\ \text{Var}(u_t) &= \mathbb{E}\left[\|u_t\|_2^2\right] = \mathbb{E}\left[x^\top W_t^\top W_t x\right]. \end{aligned} \tag{B.4.9}$$

Condition on $W_t$ and use $\mathbb{E}[xx^\top] = \Sigma_x$:

$$\begin{aligned} \mathbb{E}\left[\|u_t\|_2^2 \mid W_t\right] &= \text{tr}\left(W_t^\top W_t \Sigma_x\right) \\ &\le \|\Sigma_x\|_2\text{tr}(W_t^\top W_t) \\ &= \|\Sigma_x\|_2\|W_t\|_F^2. \end{aligned} \tag{B.4.10}$$

Decomposing $\|W_t\|_F^2$ into mean and variance terms,

$$\mathbb{E}\left[\|W_t\|_F^2\right] = \mathbb{E}\left[\|\widetilde{W}_t + \bar{W}_t\|_F^2\right] = \mathbb{E}\left[\|\widetilde{W}_t\|_F^2\right] + \|\bar{W}_t\|_F^2 = \text{Var}(W_t) + \|\mathbb{E}[W_t]\|_F^2, \tag{B.4.11}$$

where the cross term vanishes since $\mathbb{E}[\widetilde{W}_t] = 0$. Therefore,

$$\text{Var}(u_t) \le \|\Sigma_x\|_2\left(\text{Var}(W_t) + \|\mathbb{E}[W_t]\|_F^2\right), \tag{B.4.12}$$

By $\mathbb{E}[G_t] = 0$, the mean recursion (B.4.2) becomes

$$\mathbb{E}[W_{t+1}] = (1-\eta\lambda)\mathbb{E}[W_t], \tag{B.4.13}$$

hence

$$\|\mathbb{E}[W_t]\|_F^2 = (1-\eta\lambda)^{2t}\|\mathbb{E}[W_0]\|_F^2. \tag{B.4.14}$$

From the assumption in Theorem 3, $\text{Var}(G_t) \le \sigma_G^2$,

$$\text{Var}(W_t) \le (1-\eta\lambda)^{2t}\text{Var}(W_0) + \frac{\eta^2\sigma_G^2}{1-(1-\eta\lambda)^2}. \tag{B.4.15}$$

Substituting (B.4.15) and (B.4.14) into (B.4.12), we obtain

$$
\begin{aligned}
\mathrm{Var}(u_t) &\leq \|\Sigma_x\|_2 \left( (1-\eta\lambda)^{2t}\Big(\mathrm{Var}(W_0) + \|\mathbb{E}[W_0]\|_F^2\Big) + \frac{\eta^2\sigma_G^2}{1-(1-\eta\lambda)^2} \right) \\
&= \|\Sigma_x\|_2 \left( (1-\eta\lambda)^{2t}\Big(\mathrm{Var}(W_0) + \|\mathbb{E}[W_0]\|_F^2\Big) + \frac{\eta\sigma_G^2}{\lambda(2-\eta\lambda)} \right) \\
&= O\left( (1-\eta\lambda)^{2t} + \frac{\eta\sigma_G^2}{\lambda} \right),
\end{aligned}
\tag{B.4.16}
$$

where the last step uses $0 < \eta\lambda \leq 1$ so that $2 - \eta\lambda = \Theta(1)$.

The bound is the sum of two terms with different decay behaviors in $\lambda$. First, $(1-\eta\lambda)^{2t}$ decreases monotonically in $\lambda$ and captures exponential contraction of the initialization contribution under weight decay. Second, $\frac{\eta\sigma_G^2}{\lambda}$ also decreases in $\lambda$ and corresponds to the steady-state variance induced by gradient noise. Therefore, increasing $\lambda$ tightens both parts of the upper bound, implying smaller $\mathrm{Var}(u_t)$ under the stated assumptions.

$\square$

## B.5. Sequence Length Variance Control: Theory and Proof

**Theorem 4** (Sequence length reduces attention-output variance). *Assume the attention output is the uniform average* $x = \frac{1}{T}\sum_{i=1}^{T} h_i$, *where* $\{h_i\}_{i=1}^T$ *are scalar random variables (e.g., one fixed coordinate of the value vectors) that are zero-mean and mutually independent, with* $\mathrm{Var}(h_i) = \sigma^2$ *for all* $i$. *Then* $\mathrm{Var}(x) = \frac{\sigma^2}{T}$.

*Proof.* Let $x = \sum_{i=1}^{T} a_i h_i$ with $a_i \geq 0$ and $\sum_{i=1}^{T} a_i = 1$. Assume $\{h_i\}_{i=1}^T$ are independent and $\mathbb{E}[h_i] = 0$ for all $i$, with $\mathrm{Var}(h_i) = \sigma^2$. Then $\mathbb{E}[x] = 0$ and

$$
\begin{aligned}
\mathrm{Var}(x) = \mathbb{E}[x^2] &= \mathbb{E}\left[ \left( \sum_{i=1}^{T} a_i h_i \right)^2 \right] \\
&= \sum_{i=1}^{T} a_i^2 \mathbb{E}[h_i^2] + 2\sum_{1\leq i < j \leq T} a_i a_j \mathbb{E}[h_i h_j].
\end{aligned}
\tag{B.5.1}
$$

By independence and $\mathbb{E}[h_i] = 0$, for $i \neq j$ we have

$$
\mathbb{E}[h_i h_j] = \mathbb{E}[h_i]\mathbb{E}[h_j] = 0,
\tag{B.5.2}
$$

so all cross terms vanish. Also $\mathbb{E}[h_i^2] = \mathrm{Var}(h_i) = \sigma^2$. Hence,

$$
\mathrm{Var}(x) = \sum_{i=1}^{T} a_i^2 \sigma^2 = \sigma^2 \sum_{i=1}^{T} a_i^2.
\tag{B.5.3}
$$

If $a_i = 1/T$, then $\sum_{i=1}^{T} a_i^2 = 1/T$, giving $\mathrm{Var}(x) = \sigma^2/T$. $\square$

Theorem 4 captures only an averaging effect under uniform independent values. It does not by itself prove that longer-context training produces sparse attention; that claim is evaluated empirically through thresholded sparsity and entropy measurements.

## B.6. Mixture of Experts Variance Analysis: Theory and Proof

We formally characterize how Top-$k$ MoE routing reduces variance through selective averaging.

We write the gating-based MoE update in Top-$k$ set notation. Let $S(x) := \mathrm{Top}\text{-}k(\mathbf{W}_g x)$ be the selected experts, so

$$
\mathrm{MoE}(x) = \sum_{i\in S(x)} g_i(x)\mathrm{FFN}_i(x), \qquad |S(x)| = k.
\tag{B.6.1}
$$

In the uniform-gating case $g_i(x) = 1/k$ (for $i \in S(x)$), this reduces to $\text{MoE}(x) = \frac{1}{k} \sum_{i \in S(x)} \text{FFN}_i(x)$, which is the form used in Theorem 5. These assumptions are intentionally idealized. In trained MoE layers, routing is input-dependent, gates are unequal, and experts are statistically coupled through joint training; hence the $1/k$ factor should be read as a limiting intuition rather than a predictive law.

**Theorem 5** (Top-$k$ MoE reduces layer-output and Jacobian variance). *Fix a Top-$k$ MoE layer with uniform gating*

$$\text{MoE}(x) = \frac{1}{k} \sum_{i \in S(x)} \text{FFN}_i(x), \qquad |S(x)| = k, \tag{B.6.2}$$

*and assume the routing set $S(x)$ is locally constant around $x$.*

*(Output variance). If $\{\text{FFN}_i(x)\}_{i \in S(x)}$ are mutually independent with common mean and variance, then*

$$\text{Var}\left[\text{MoE}(x)\right] = \frac{1}{k} \text{Var}\left[\text{FFN}_1(x)\right]. \tag{B.6.3}$$

*(Jacobian variance). Let $J_i(x) := \frac{\partial \text{FFN}_i(x)}{\partial x}$ and $J^{\text{MoE}}(x) := \frac{\partial \text{MoE}(x)}{\partial x} = \frac{1}{k} \sum_{i \in S(x)} J_i(x)$. If $\{J_i(x)\}_{i \in S(x)}$ are mutually independent with common mean and variance, then*

$$\text{Var}\left[J^{\text{MoE}}(x)\right] = \frac{1}{k} \text{Var}\left[J_1(x)\right]. \tag{B.6.4}$$

Theorem 5 shows that Top-$k$ MoE reduces variance by an averaging effect: under uniform gating, the MoE output and Jacobian are averages over $k$ selected experts, and under independence the variance decreases by a factor $1/k$.

*Proof.* First we want to prove the **output variance** in Equation (B.6.3). Let $u_i(x_\ell) := \text{FFN}_i(x_\ell) \in \mathbb{R}^d$ denote the $i$-th expert output, and

$$u^{\text{MoE}}(x_\ell) := \text{MoE}(x_\ell) = \frac{1}{k} \sum_{i \in \text{Top-}k} u_i(x_\ell). \tag{B.6.5}$$

By the assumption that the selected expert outputs $\{u_i(x_\ell)\}_{i \in \text{Top-}k}$ are mutually independent, with common mean $\mathbb{E}[u_i(x_\ell)] = m_\ell$ and common second moment, then $\mathbb{E}[u^{\text{MoE}}(x_\ell)] = m_\ell$, and

$$u^{\text{MoE}}(x_\ell) - m_\ell = \frac{1}{k} \sum_{i \in \text{Top-}k} \left(u_i(x_\ell) - m_\ell\right). \tag{B.6.6}$$

Therefore,

$$\begin{aligned}
\text{Var}\left[u^{\text{MoE}}(x_\ell)\right] &= \mathbb{E}\left[\left\|\frac{1}{k} \sum_{i \in \text{Top-}k} \left(u_i(x_\ell) - m_\ell\right)\right\|_2^2\right] \\
&= \frac{1}{k^2} \mathbb{E}\left[\left\|\sum_{i \in \text{Top-}k} \left(u_i(x_\ell) - m_\ell\right)\right\|_2^2\right].
\end{aligned} \tag{B.6.7}$$

Expanding the Equation (B.6.7) yields

$$\left\|\sum_{i \in \text{Top-}k} \left(u_i(x_\ell) - m_\ell\right)\right\|_2^2 = \sum_{i \in \text{Top-}k} \|u_i(x_\ell) - m_\ell\|_2^2 + 2 \sum_{\substack{i,j \in \text{Top-}k \\ i<j}} \langle u_i(x_\ell) - m_\ell, u_j(x_\ell) - m_\ell \rangle. \tag{B.6.8}$$

Using independence with zero-mean centered terms gives

$$\mathbb{E}\left[\langle u_i(x_\ell) - m_\ell, u_j(x_\ell) - m_\ell \rangle\right] = \langle \mathbb{E}[u_i(x_\ell) - m_\ell], \mathbb{E}[u_j(x_\ell) - m_\ell] \rangle = 0, \qquad (i \neq j), \tag{B.6.9}$$

so all cross terms vanish. Hence

$$
\begin{aligned}
\mathrm{Var}\left[u^{\mathrm{MoE}}(x_\ell)\right] &= \frac{1}{k^2} \sum_{i \in \text{Top-}k} \mathbb{E}\left[\|u_i(x_\ell) - m_\ell\|_2^2\right] \\
&= \frac{1}{k^2} \cdot k \mathrm{Var}\left[u_1(x_\ell)\right] \\
&= \frac{1}{k} \mathrm{Var}\left[\mathrm{FFN}_1(x_\ell)\right].
\end{aligned}
\tag{B.6.10}
$$

Next we want to prove the **Jacobian variance** in Equation (B.6.4). For the variance, in a standard feed-forward network (FFN), the Jacobian variance at layer $\ell$ is

$$
\mathrm{Var}\left[J_\ell\right] = \mathrm{Var}\left(\frac{\partial \mathrm{FFN}(x_\ell)}{\partial x_\ell}\right).
\tag{B.6.11}
$$

For a mixture-of-experts (MoE) layer with Top-$k$ routing, the output is

$$
\mathrm{MoE}(x_\ell) = \frac{1}{k} \sum_{i \in \text{Top-}k} \mathrm{FFN}_i(x_\ell),
\tag{B.6.12}
$$

with Jacobian

$$
J_\ell^{\mathrm{MoE}} = \frac{\partial \mathrm{MoE}(x_\ell)}{\partial x_\ell} = \frac{1}{k} \sum_{i \in \text{Top-}k} \frac{\partial \mathrm{FFN}_i(x_\ell)}{\partial x_\ell} = \frac{1}{k} \sum_{i \in \text{Top-}k} J_{\ell,i}.
\tag{B.6.13}
$$

Assume the selected experts have mutually independent Jacobians and identical second-moment structure, we have

$$
\mathrm{Var}\left[J_\ell^{\mathrm{MoE}}\right] = \mathrm{Var}\left(\frac{1}{k} \sum_{i \in \text{Top-}k} J_{\ell,i}\right) = \frac{1}{k^2} \sum_{i \in \text{Top-}k} \mathrm{Var}\left[J_{\ell,i}\right].
\tag{B.6.14}
$$

If each expert has the same variance as a standard FFN, then

$$
\mathrm{Var}\left[J_\ell^{\mathrm{MoE}}\right] = \frac{1}{k} \mathrm{Var}\left[J_\ell^{\mathrm{FFN}}\right].
\tag{B.6.15}
$$

Thus, under these assumptions, the MoE layer introduces a variance reduction factor $1/k$ compared with a single FFN layer. $\qquad\square$

## B.7. MoE Jacobian Norm Scaling under Uniform Top-$k$ Routing

By (Sun et al., 2025), we can estimate the upper bound of the gradient norm of $\frac{\partial y_\ell}{\partial x_1}$:

$$
\left\|\frac{\partial y_\ell}{\partial x_\ell'}\right\|_2 \le 1 + \left\|\frac{\partial \mathrm{FFN}(\mathrm{LN}(x_\ell'))}{\partial \mathrm{LN}(x_\ell')}\right\|_2 \left\|\frac{\partial \mathrm{LN}(x_\ell')}{\partial x_\ell'}\right\|_2.
\tag{B.7.1}
$$

For MoE systems (Shazeer et al., 2017), based on Equation (B.7.1), we have

$$
\begin{aligned}
y_\ell &= x_\ell' + \mathrm{MoE}\left(\mathrm{LN}(x_\ell')\right) \\
&= x_\ell' + \sum_{i \in \text{Top-}k} g_i \mathrm{FFN}_i\left(\mathrm{LN}(x_\ell')\right) \\
&= x_\ell' + \frac{1}{k} \sum_{i \in \text{Top-}k} \mathrm{FFN}_i\left(\mathrm{LN}(x_\ell')\right), \qquad \left(\text{since } g_i = \frac{1}{k}\right).
\end{aligned}
\tag{B.7.2}
$$

Assuming the routing set Top-$k$ is locally constant around $x_\ell'$, the Jacobian satisfies

$$
\begin{aligned}
\left\|\frac{\partial y_\ell}{\partial x_\ell'}\right\|_2 &= \left\|I + \frac{1}{k} \sum_{i \in \text{Top-}k} \frac{\partial \mathrm{FFN}_i(\mathrm{LN}(x_\ell'))}{\partial x_\ell'}\right\|_2 \\
&\le 1 + \frac{1}{k} \left\|\sum_{i \in \text{Top-}k} \frac{\partial \mathrm{FFN}_i(\mathrm{LN}(x_\ell'))}{\partial x_\ell'}\right\|_2.
\end{aligned}
\tag{B.7.3}
$$

We know that,

$$\frac{\partial \mathrm{FFN}_i(\mathrm{LN}(x'_\ell))}{\partial x'_\ell} = \frac{\partial \mathrm{FFN}_i(\mathrm{LN}(x'_\ell))}{\partial \mathrm{LN}(x'_\ell)} \frac{\partial \mathrm{LN}(x'_\ell)}{\partial x'_\ell}, \tag{B.7.4}$$

hence

$$\left\| \frac{\partial y_\ell}{\partial x'_\ell} \right\|_2 \leq 1 + \frac{1}{k} \left\| \sum_{i \in \mathrm{Top}\text{-}k} \frac{\partial \mathrm{FFN}_i(\mathrm{LN}(x'_\ell))}{\partial \mathrm{LN}(x'_\ell)} \right\|_2 \left\| \frac{\partial \mathrm{LN}(x'_\ell)}{\partial x'_\ell} \right\|_2. \tag{B.7.5}$$

After adding a scaling factor, we obtain

$$\left\| \frac{\partial y_\ell}{\partial x'_\ell} \right\|_2 \leq 1 + \frac{1}{k} \left\| \sum_{i \in \mathrm{Top}\text{-}k} \frac{\partial \mathrm{FFN}_i(\mathrm{LN}(x'_\ell))}{\partial \mathrm{LN}(x'_\ell)} \right\|_2 \left\| \frac{1}{\sqrt{\ell}} \right\|_2 \left\| \frac{\partial \mathrm{LN}(x'_\ell)}{\partial x'_\ell} \right\|_2. \tag{B.7.6}$$

For layers with MoE-FFN, we apply this MoE-specific scaling factor; for layers other than MoE-FFN, we use the original scaling factor.

## C. Experimental Settings

Across all our experiments, we implement a standard LLM transformer architecture with RoPE (Su et al., 2024), SwiGLU (Shazeer, 2020), and RMSNorm (Zhang and Sennrich, 2019). We utilize the FineWeb-Edu dataset[1] for training models across all our experiments and tokenize the dataset with the GPT-NeoX tokenizer[2]. Although different experiment groups may utilize different data sizes, all our evaluation perplexities are computed on the same held-out evaluation set from FineWeb-Edu. In our experiments, we adapt the training framework from OLMo[3] and use the AdamW optimizer with mixed precision training. All our experiments are run on Hopper-series GPUs.

### C.1. Depth Control Experiments

*Table 8.* Model configuration for Section 2.

| Depth $L$ | # of Param. | Hidden Dim | MLP Hidden Dim | Query Head | Key-Value Head | Max Training Length |
|---|---|---|---|---|---|---|
| L=12 | 900M | | | | | |
| L=16 | 1.2B | | | | | |
| L=24 | 1.7B | 2048 | 8192 | 16 | 16 | 1024 |
| L=32 | 2.3B | | | | | |

In Section 2, we vary the model depths to verify the phenomenon of CoD. Specific details about each model are in Table 8. In all runs, we keep the weight decay at 0.1 and warmup iterations at 1000 with a warmup cosine learning rate scheduler. We sweep the learning rate over $\{5 \times 10^{-5}, 1 \times 10^{-4}, 5 \times 10^{-4}, 8 \times 10^{-4}, 1 \times 10^{-3}\}$ and choose $1 \times 10^{-3}$ for $L \in \{12, 16, 24\}$, and $8 \times -4$ for $L = 32$. All runs are trained on 10B tokens with global batch size of 256, and we evaluate the downstream performance on MMLU, ARC-Easy, ARC-Challenge, and Hellaswag using the implementation from lm-eval-harness[4] with zero-shot settings, reporting accuracy for MMLU and normalized accuracy (acc_norm) for other tasks.

### C.2. Weight Decay Experiments

All of the experiments in Section 4.1.1 are trained with a model of $L = 16$ layers, $d = 2048$ hidden dimensions, MLP dimension of 8192, and $h = 16$ attention heads (with the same number of key-value heads). We vary the weight decay strength of the AdamW optimizer and use the same learning rate of $1 \times 10^{-3}$, 1000 warmup iterations, and a warmup cosine learning rate scheduler. All models are trained on 5B tokens with a global batch size of 128.

---

[1]https://huggingface.co/datasets/HuggingFaceFW/fineweb-edu
[2]https://github.com/EleutherAI/gpt-neox
[3]https://github.com/allenai/OLMo
[4]https://github.com/EleutherAI/lm-evaluation-harness

## C.3. Sequence Length Experiments

All of the experiments regarding sequence length in Section 4.1.2 are trained with a model of $L = 16$ layers, $d = 2048$ hidden dimensions, MLP dimension of 8192, and $h = 16$ attention heads (with the same number of key-value heads). We keep the weight decay at 0.1, learning rate at $1 \times 10^{-3}$, 1000 warmup iterations, and a warmup cosine learning rate scheduler. All models are trained on 5B tokens with a global batch size of 256.

## C.4. Group Query Attention Experiments

The experiments regarding GQA in Section 4.2.1 are trained with a model of $L = 16$ layers, $d = 2048$ hidden dimensions, MLP dimension of 8192, and $h = 16$ attention heads. The group size settings $G \in \{1, 4, 16\}$ correspond to key-value head sizes of 16, 4, and 1, respectively. With this configuration, the model sizes are approximately 1.2B (1.176B), 1.1B (1.076B), and 1.051B parameters. We keep the training sequence length at 4096, weight decay at 0.1, learning rate at $1 \times 10^{-3}$, 1000 warmup iterations, and a warmup cosine learning rate scheduler. The $G = 1$ models are trained on 5B tokens, while $G = 4$ and $G = 16$ are trained on 5.35B and 5.6B tokens, respectively, to keep the training FLOPs constant. All experiments are trained with a global batch size of 256.

## C.5. Mixture of Expert Experiments

*Table 9.* Model configuration for Section 4.2.2.

| | Depth | Hidden Dim | MLP Hidden Dim | Query Head | Key-Value Head | Total Experts | Activated Experts | Shared Experts |
|---|---|---|---|---|---|---|---|---|
| Dense-400M | | 1536 | 3072 | 12 | | - | - | - |
| MoE-2BA400M | 16 | 768 | 1536 | 12 | | 32 | 4 | 1 |
| Dense-1B | | 2048 | 8192 | 16 | | - | - | - |
| MoE-7BA1B | | 2048 | 2048 | 16 | | 64 | 8 | - |

We adopt the kernel and MoE implementation from OLMo-core[5] and utilize the implementation of DropLess MLP (Gale et al., 2022). For both size configurations of MoE training, we set the load balancing loss (Shazeer et al., 2017) to 0.01 and router z-loss to 0.001 (Zoph et al., 2022) according to the practice of OLMoE (Muennighoff et al., 2024). The detailed configurations of the models for the two setups are presented in Table 9. For all runs, we set the training sequence length at 4096, weight decay at 0.1, and 1000 warmup iterations with a warmup cosine learning rate scheduler, training on 5B total tokens with a global batch size of 256. We set the learning rate to $1 \times 10^{-3}$ for Dense-400M, MoE-2BA400M, and Dense-1B experimental runs, but set the learning rate for MoE-7BA1B to $4 \times 10^{-4}$ according to learning rate sweep results.

## C.6. Ablation Experiments

*Table 10.* Model configuration for Section 5

| | Depth | Hidden Dim | MLP Hidden Dim | Query Head | Key-Value Head | Total Experts | Activated Experts | Shared Experts |
|---|---|---|---|---|---|---|---|---|
| $L = 16$ | 16 | 2048 | 8192 | | 16 | - | - | - |
| $L = 32$ | 32 | 1536 | 6144 | | 16 | - | - | - |
| MoE-$L = 32$ | 32 | 1120 | 1120 | | 16 | 64 | 8 | - |

For all experiments in Section 5, we use 1.2B parameter models with configurations detailed in Table 10. We train on 10B tokens from FineWeb-Edu and report evaluation perplexity on a held-out dataset, along with zero-shot accuracy on ARC-Challenge (Clark et al., 2018) and HellaSwag (Zellers et al., 2019).

All experiments use the AdamW optimizer with a global batch size of 256. Unless otherwise specified, models are trained with weight decay of 0.1, context length of 1024, 1000 warmup iterations, and a cosine learning rate scheduler. We use a learning rate of $4 \times 10^{-4}$ for MoE models and $1 \times 10^{-3}$ for all other configurations.

---

[5]https://github.com/allenai/OLMo-core

## C.7. Weight Decay

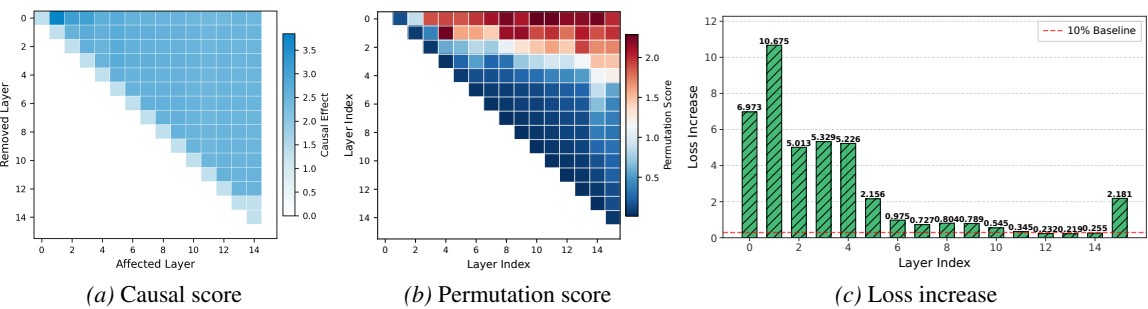

*(a)* Causal score      *(b)* Permutation score      *(c)* Loss increase

*Figure 20.* Score visualziation for weight decay $\lambda = 0.1$.

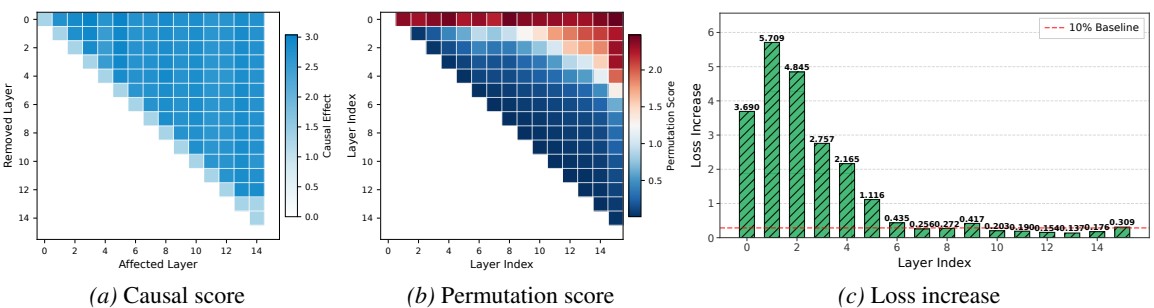

*(a)* Causal score      *(b)* Permutation score      *(c)* Loss increase

*Figure 21.* Score visualziation for weight decay $\lambda = 1$.

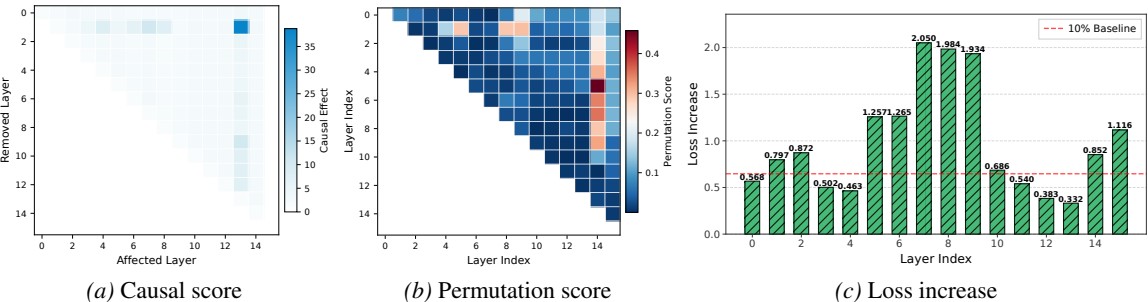

*(a)* Causal score      *(b)* Permutation score      *(c)* Loss increase

*Figure 22.* Score visualziation for weight decay $\lambda = 3$.

Visualizations of each score for $\lambda \in \{0.1, 1, 3\}$ are provided in Figures 20 to 22, respectively.

## C.8. Sequence Length

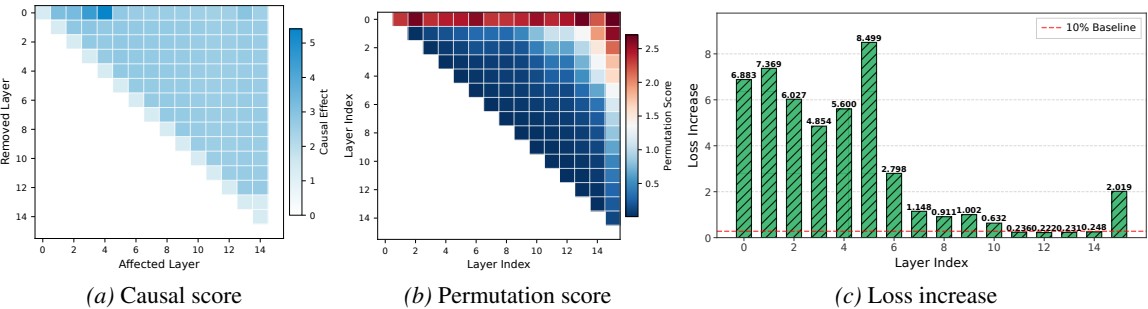

*(a)* Causal score      *(b)* Permutation score      *(c)* Loss increase

*Figure 23.* Score visualziation for sequence length $T = 1024$.

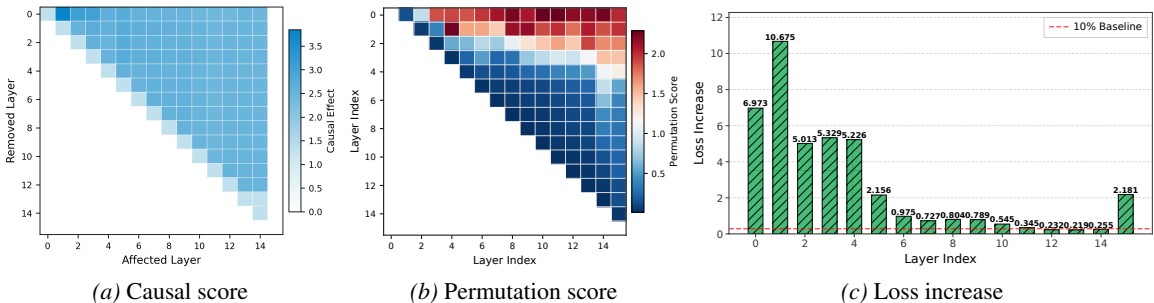

*(a)* Causal score      *(b)* Permutation score      *(c)* Loss increase

*Figure 24.* Score visualziation for sequence length $T = 4096$.

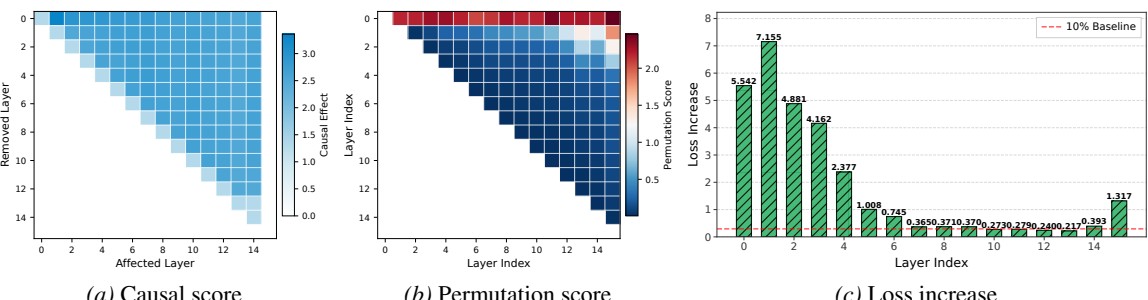

*(a)* Causal score      *(b)* Permutation score      *(c)* Loss increase

*Figure 25.* Score visualziation for sequence length $T = 8192$.

Visualizations of each score for $T \in \{1024, 4096, 8192\}$ are provided in Figures 23 to 25, respectively.

## C.9. Grouped Query Attention

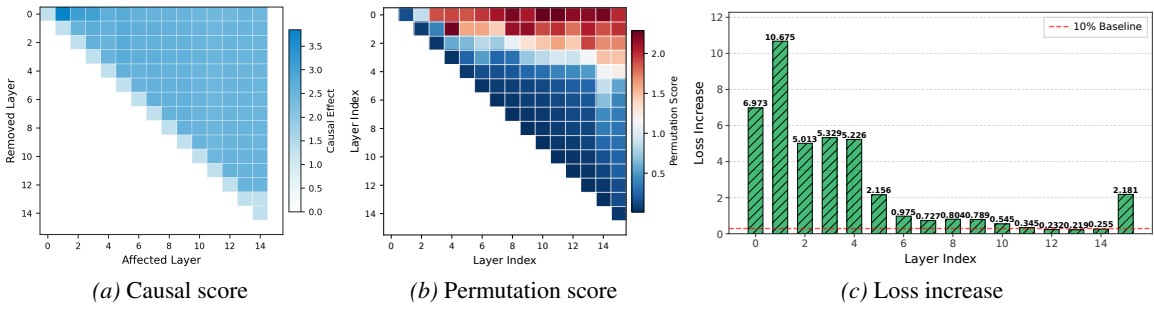

*(a)* Causal score      *(b)* Permutation score      *(c)* Loss increase

*Figure 26.* Score visualziation for MHA $G = 1$.

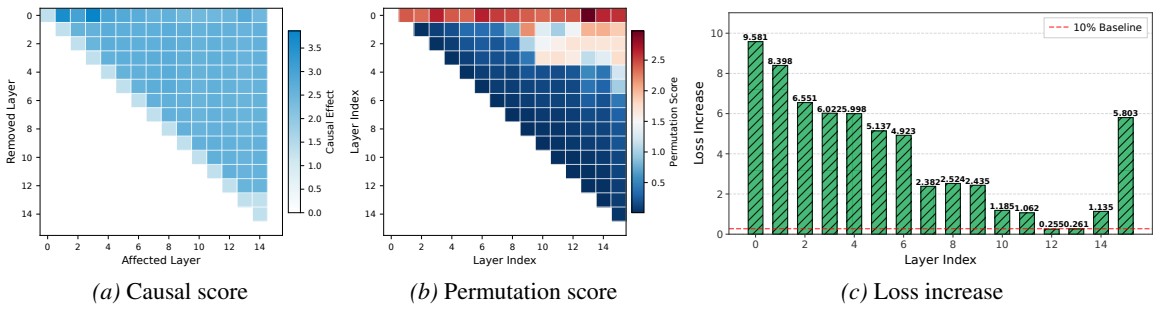

*(a)* Causal score      *(b)* Permutation score      *(c)* Loss increase

*Figure 27.* Score visualziation for MQA $G = 16$.

Visualizations of each score for $G \in \{1, 16\}$ are provided in Figures 26 and 27, respectively.

## C.10. Mixture of Experts

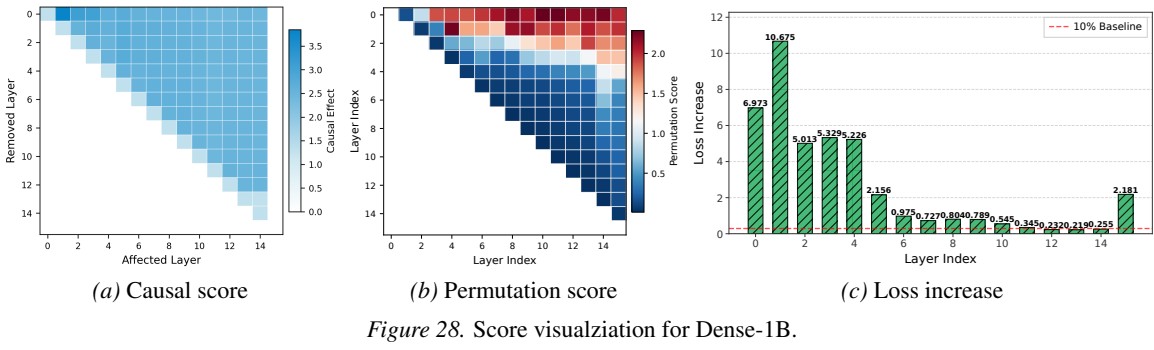

*(a)* Causal score          *(b)* Permutation score          *(c)* Loss increase

*Figure 28.* Score visualziation for Dense-1B.

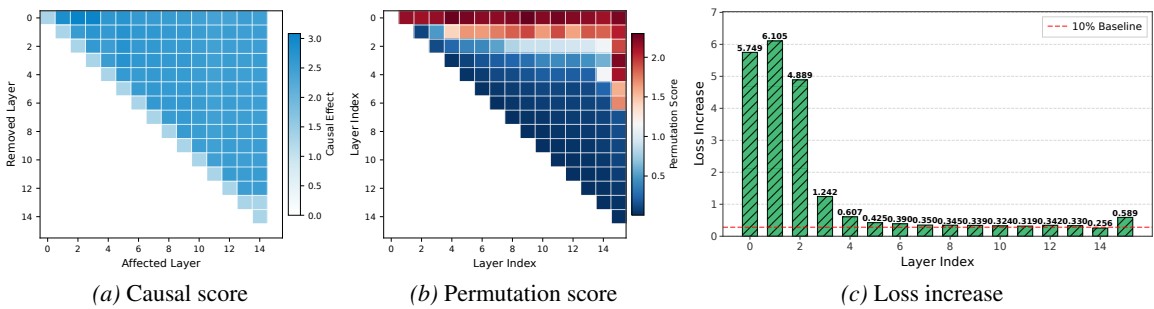

*(a)* Causal score          *(b)* Permutation score          *(c)* Loss increase

*Figure 29.* Score visualziation for MoE-7BA1B.

Visualizations of Dense-1B and MoE-7BA1B are provided in Figures 28 and 29, respectively.

