# OpenReview forum: "When Does Sparsity Mitigate the Curse of Depth in LLMs"
_ICML.cc/2026/Conference — ICML 2026 regular_

### Official Review · Reviewer_9xNW · 2026-03-10

**Soundness:** 2
**Presentation:** 3
**Significance:** 4
**Originality:** 4
**Overall Recommendation:** 3
**Confidence:** 4

**Summary:**

In this paper, the authors argue that sparsity acts as a regulator of variance propagation, thereby mitigating the curse of depth in Large Language Models (LLMs). To validate this hypothesis, the authors use Theorem 1 to mathematically establish the relationship between sparsity and deep residual variance. Furthermore, they experimentally analyse sparsity across two distinct dimensions: implicit sparsity (such as weight and attention sparsity) and explicit sparsity (such as Grouped-Query Attention and Mixture-of-Experts).

**Compliance With Llm Reviewing Policy:**

Affirmed.

**Final Justification:**

I am maintaining my score, as the authors have not fully addressed my concerns. For example, in Q2, there is a serious contradiction in their work: whilst they argue that highly sparse attention mitigates the 'curse of depth', their theoretical proofs invalidate this claim by inexplicably relying on the assumption of a perfectly uniform ($1/n$) attention distribution.

**Key Questions For Authors:**

1. Could you clarify or revise the idealised assumptions underlying the theorems to better reflect actual training dynamics?
2. Could you explain the discrepancy between the theoretical use of $\rho_l$ and the proxy metrics used in your experiments?
3. Could you discuss how these results would hold up at a true LLM scale?

**Limitations:**

Yes.

**Strengths And Weaknesses:**

Strengths
1. Investigating the intrinsic link between sparsity and the curse of depth (variance propagation) offers a highly relevant and interesting research direction.
2. Alongside the empirical results, the paper provides mathematical results to describe how the degree of sparsity directly influences deep residual variance.

Weaknesses
1. Idealised Theoretical Assumptions:
The assumptions underpinning the theoretical analysis are too idealised and heavily disconnected from actual network dynamics. For instance, Theorem 1 assumes that for each layer $l$, the weight matrix $W_l$ is independent of the input and mask $(r_l, D_l)$. This assumption only practically holds at random initialisation. In fully trained LLMs, $W_l$ is updated via backpropagation based on the input $r_l$, meaning they are highly coupled and exhibit strong correlation.
Similarly, Theorem 2 introduces several constants (e.g., $\kappa_{l,h}$, $\nu_{l,h}$, and $\omega_{l}$)  and assumes they hold uniformly across all inputs $X$. These conditions are practically impossible to satisfy during actual training. Consequently, Theorem 2 reads more like an existence proof rather than actionable theoretical guidance, as it offers no methodology for estimating or controlling these constants in practice.
2. Discrepancy Between Theory and Experiments:
The theoretical section heavily focuses on the impact of the mask density parameter $\rho_l$ on variance , yet the experimental section relies on metrics like sequence length, weight decay, GQA, and MoE to indirectly represent sparsity. The authors neither mathematically prove that these mechanisms directly reduce $\rho_l$, nor do they empirically measure any changes in $\rho_l$ during training.
3. Insufficient Experimental Scale:
The experimental setup falls significantly short of standard LLM scales, raising valid concerns about the generalisability of the findings. Despite the title claiming to study 'LLMs', the experiments evaluate models ranging only from 900M to 2.3B parameters , with a maximum depth of 32 layers. Furthermore, the pre-training corpus is limited to 5B to 10B tokens. By contrast, mainstream open-weight LLMs are typically trained on trillions of tokens and feature 80 or more layers. Thus, results at a true LLM scale are needed to verify that these conclusions generalise.

Typos & Formatting Issues
(1) In Theorem 1, variables are improperly included inside the Big-O notation (e.g., $\mathcal{O}((1+\sqrt{\alpha\rho})^{2L})$).
(2) Several references are incomplete or poorly formatted. For example, the citation 'A Vaswani. Attention is all you need. NeurIPS, 2017' is missing the full author list.

---

> ### Author Rebuttal · Authors · 2026-03-31
>
> We sincerely thank the reviewer for the careful reading and positive evaluation. We are glad the link between sparsity and variance propagation, and the breadth of experiments over implicit/explicit sparsity, were recognized. We address the concerns below.
>
> **Q1. Could you clarify or revise the idealized assumptions underlying the theorems to better reflect actual training dynamics?**
>
> **A1.** Thank you for this comment. We would like to clarify that our theoretical analysis is structural rather than training-dynamic. In the literature on neural networks, these are two lines of study. One studies properties implied by the network architecture itself, while the other studies how individual weights evolve during optimization, often from initialization to the end of training [1].
>
> Our paper belongs to the first category. The theorems analyze how network structure affects variance propagation across depth. Thus the assumptions are on the layerwise operator form and effective sparsity pattern, not on the optimization trajectory. The property is intended to hold for the network mapping at any training epoch, since it comes from the architecture-level update form rather than a specific model of learning dynamics [1,2,3]. By contrast, training-dynamics analyses require assumptions on gradient flow, parameter evolution, initialization, optimizer behavior, or the final weight distribution [2]. That is a different question. Hence our theorems do not assume training dynamics; they assume independence and simplifications of the Transformer structure.
>
> [1] Boris Hanin et al., "The Principles of Deep Learning Theory".
>
> [2] Andrew M. et al., "Exact solutions to the nonlinear dynamics of learning in deep linear neural networks."
>
> [3] Arthur Jacot et al., "Neural Tangent Kernel."
>
> **Q2. Could you explain the discrepancy between the theoretical use of sparsity and the proxy metrics used in your experiments?**
>
> **A2.** Thank you for the comment. In theory, sparsity is represented by mask $D_\ell$ and effective density $\rho_\ell$, which measures how much input signal passes through the layer update; this is what enters the variance bound. In experiments, we use mechanism-specific proxies for the same quantity: weight sparsity by the fraction of near-zero weights, attention sparsity by the fraction of near-zero attention entries, GQA by the degree of KV sharing, and MoE by the number of activated experts. The appendix gives the detailed mappings: weight decay in Sec. B.4, sequence length in Sec. B.5, MHA/GQA in Sec. B.6, and MoE in Sec. B.7. Thus, the definitions in the theory are fully aligned with the experiments, while the main theory for sparsity is a general definition.
>
> We do explicitly measure the corresponding sparsity quantities in the experiments through these mechanism-specific proxies. What we do not claim is that the main theorem separately derives, from first principles, how each mechanism reduces $\rho_\ell$; rather, the theorem is stated at the level of a general effective sparsity parameter, and the appendix shows how each practical mechanism maps to it.
>
> **Q3. Could you discuss how these results would hold up at a true LLM scale?**
>
> **A3.** We first note that the curse of depth has already been documented in larger open-source LLMs [4,5], which motivates our study. In early experiments, we also trained a 1.2B dense model ($L=16$) with 30B tokens (already exceeding Chinchilla-optimal) and a 7B model ($L=32$) with 10B tokens under different sequence lengths. In both, longer sequence length consistently gave lower peak variance, higher usefulness, and better final training loss, motivating the systematic study in this paper.
> | Model Size | Seq. Len. | Data Size | Peak Var. @ Last Layer | Usefulness Score | Train Loss |
> | - | - | - | - | - | - |
> | 1.2B | 512 | 30B | $\approx$ 7e3 | 0.69 | 2.78 |
> | 1.2B | 4096 | 30B | $\approx$ 700 | 0.81 | 2.63 |
> | 7B | 512 | 10B | $\approx$ 1e5 | 0.63 | 2.64 |
> | 7B | 4096 | 10B | $\approx$ 3e4 | 0.75 | 2.50 |
>
> More broadly, modern LLMs are increasingly adopting GQA, MoE, and long-context pretraining. We argue this is not only for efficiency, but also because the sparsity induced by these mechanisms helps the network use depth more effectively by reducing identity-mapping layers.
>
> Regarding experimental scale: even in mainstream open-source LLM development, the standard practice is to first verify findings at small scale before extending to larger scale. Unfortunately, the latter phase requires trillion-token training runs that are impractical for academic research. Given our compute budget, we chose the 1-2B scale to provide the most thorough and systematic ablations possible. We view this work as a strong first phase that establishes the theoretical and empirical basis for future verification at larger scale.
>
> [4] Csordás, et al. "Do language models use their depth efficiently?.
>
> [5] Sun, et al. "The curse of depth in large language models."

---

> > ### Author Rebuttal · Reviewer_9xNW · 2026-04-03
> >
> > Thank you for your responses to my initial concerns. I have some additional questions regarding your reply.
> > 1. Q1
> > In Theorem 1, you assume that for each layer, the weight matrix is independent of the input and the mask. In reality, this assumption breaks down almost immediately after the very first pass of backpropagation. In your rebuttal, you stated that 'The property is intended to hold for the network mapping at any training epoch'. However, in the wider theoretical literature, this sort of independence assumption is typically made purely for mathematical convenience, rather than expecting it to hold strictly true throughout the training process. Could you please clarify your stance on this? Are you claiming this independence genuinely persists across all epochs, or acknowledging it as a necessary theoretical simplification?
> > Furthermore, in Section 3.2.1 of your paper, you discuss 'implicit sparsity', explicitly describing it as being 'induced dynamically during training'. If implicit sparsity emerges as a direct result of the training process, which fundamentally relies on the deep, continuous coupling between weights and inputs, how can its mechanism be explained by Theorem 1, a theorem entirely built on the assumption that weights and inputs are completely independent? Could you please clarify this?
> >
> > 2. Q2
> > Appendices B.4 to B.7 fail to establish valid mathematical mapping between your practical mechanisms (weight decay, sequence length, GQA, MoE) and your core theoretical parameter, $\rho_\ell$ (mask density). In fact, what you have demonstrated is that, under highly simplified assumptions, each mechanism reduces its own output variance. Your appendix derivations do not establish a connection between the mechanisms and $\rho_\ell$, which undermines the logic of the theoretical analysis.
> > Mathematically, your derivation in Equation 83 is questionable.
> > Firstly, your foundational assumption that each element of the attention matrix $A$ is roughly $1/n$ is unreasonable. In trained Large Language Models, attention distributions are notoriously peaked and sparse, a phenomenon your own paper explicitly relies upon when discussing 'implicit sparsity'. Assuming a perfectly uniform distribution across tokens contradicts both empirical reality and your own narrative.
> > Secondly, your subsequent algebraic derivation is incorrect.  When factoring the uniform weight $1/n$ out of the variance operator, it should be squared $(1/n^2)$. You failed to square it, leading to the incorrect derivation: $\text{Var}(\text{Attn}) \sim \frac{1}{n}\sum_{i=1}^{n}\text{Var}(V_i)d_{\text{head}} = \sigma_V^2 d$. Besides confusing $d_{\text{head}}$ and $d$, this faulty derivation concludes that variance is a constant independent of $n$, directly contradicting your own Theorem 5 conclusion ($\text{Var}(\text{Attn}) = \sigma_V^2 / n$ ).
> > 3. Q3
> > I acknowledge the additional experimental results provided in your rebuttal (the 1.2B and 7B models) and appreciate the effort invested during this process. However, this data fails to adequately address my fundamental concern regarding actual LLM scales and the curse of depth.
> > Specifically, scaling the parameter count to 7B while keeping the depth fixed at 32 layers misses the core issue of extreme depth. Validating your claims on a 32-layer model offers no rigorous guarantee that your findings will generalise to genuinely deep models of 80 layers or more.

---

> > > ### Author Response · Authors · 2026-04-04
> > >
> > > **Q1**
> > >
> > > We think the concern may come from combining two different lines of neural network theory.
> > >
> > > The first line studies training dynamics. There, one tracks how a fixed-width-depth network changes over training time $t$, from initialization toward later epochs (to $T = \infty$), and how the distribution of activations or outputs changes as the weights are updated. In that setting, the current input, hidden states, and weights are all coupled through optimization, so independence between weights and inputs generally does not hold. This is the setting studied in works on gradient flow and related optimization-based analyses [1][2].
> > >
> > > The second line studies network structure. There, time $t$ is fixed, and one analyzes how an input propagates through $L$ layers, or what happens when depth or width becomes large. In this setting, the object of study is the architecture-level mapping itself. Our paper studies the curse of depth, so it falls in this second category. For that reason, the assumptions in Theorem 1 are structural assumptions on the layerwise operator form and sparsity mechanism [3][4].
> > >
> > > So our perspective is the second: we use it as a standard analysis of variance propagation through depth at a fixed epoch $T$. This is exactly why our rebuttal said network mapping at any training epoch: at each fixed epoch, one may study the resulting architecture-level map, without modeling the full optimizer-induced dependence over time.
> > >
> > > This also helps clarify the point about Section 3.2.1 and “implicit sparsity induced dynamically during training.” Here, “induced” does not mean that Theorem 1 is a theorem about optimization dynamics or implicit bias in the training-theoretic sense. It means that, at a given epoch $T$, the trained network exhibits the sparsity pattern. So there is no contradiction.
> > >
> > > More broadly, it may be helpful not to conflate the limits in these two theories. One studies depth $L$ or width asymptotics at a fixed epoch (Sparsity and Criticality are the two most important property of neural Networks). In training-dynamics analyses, one studies evolution over time $T$. Simple mathematical analysis tells us that we can not move the two limits to $\infty$ at the same time.
> > >
> > > [1] Jacot, A., et. al. *Neural Tangent Kernel: Convergence and Generalization in Neural Networks*.
> > >
> > > [2] Chizat, L., et. al. *On Lazy Training in Differentiable Programming*.
> > >
> > > [3] Boris Hanin, *The Principles of Deep Learning Theory*.
> > >
> > > [4] Andrew M. et. al. *Exact solutions to the nonlinear dynamics of learning in deep linear neural networks*.
> > >
> > > **Q2**
> > > **A2.** We respectfully clarify that there is no discrepancy between the main theorem and Appendix B.4--B.7. In **Theorem 1**, $\rho_\ell$ is introduced as a **general sparsity parameter**. In Appendix B.4--B.7, we then give **specific instantiations** of this same general quantity for weight decay, seq. length, GQA, and MoE. The notation changes because the mechanisms are different, but the mathematical role is the same: each case specifies how many effective value remain active, which is exactly the Bernoulli-style sparsity quantity measured by the fraction of active 1 versus 0 entries. So the theorem is general, and the appendix gives its mechanism-specific specializations.
> > >
> > > For the attention assumption, we agree that taking each attention entry to be roughly $1/n$ is an idealized mean-field simplification rather than a literal description of trained LLM attention. This type of idealization is standard in Transformer theory [5][6].
> > >
> > > For Equation 83, you are correct. This is a typo, and we apologize for it. This is exactly why the later GQA expression has the extra $1/G$ factor; otherwise one would incorrectly lose an $n$ factor. **But the typo only influence this line, and the following proof is not influenced**. We will revise this typo in the next version.
> > >
> > > [5] Greg Yang, et. al. *Tensor Programs V: Tuning Large Neural Networks via Zero-Shot Hyperparameter Transfer*.
> > >
> > > [6] Sho Takase, et. al. *Spike No More: Stabilizing the Pre-training of Large Language Models*.
> > >
> > > **Q3.**
> > >
> > > **A3.** We should clarify that for a 7B-style model, **32 layers is a realistic choice rather than an artificially shallow setting**. For example, **Qwen3-8B uses 36 layers**, so our 7B/32L experiment is aligned with modern LLM design. So, this experiment is relevant for real LLM practice.
> > >
> > > To address the reviewer’s extreme-depth concern, we also ran a **7B, $L=80$** experiment (consider the limited rebuttal time window, we just run with 5B tokens).  In this regime, variance explosion is substantially worse, but the same trend remains: **sparsity-based regularization reduces last-layer variance, improves usefulness, and lowers training loss**.
> > >
> > > | Model | Seq. Len.|Data Size | Peak Var. @ Last Layer | Useful. Score | Train Loss|
> > > | - | - |- | - | - | - |
> > > | 7B$L=80$     | 1024      |5B     | $\approx$ 5e5    | 0.45 | 2.76|
> > > | 7B$L=80$     | 4096     |5B     | $\approx$ 6e4    | 0.53 | 2.67|

---

### Official Review · Reviewer_uDLV · 2026-03-12

**Soundness:** 2
**Presentation:** 2
**Significance:** 2
**Originality:** 3
**Overall Recommendation:** 3
**Confidence:** 3

**Summary:**

This paper studies whether sparsity mitigates the “curse of depth” in Pre-LN Transformers. It considers both implicit sparsity (from training/data: weight decay and long-context attention patterns) and explicit sparsity (from architecture: GQA and MoE), and argues, using a mix of theory and experiments, that more sparsity tends to reduce the model’s output variance, which improves depth usage (via layer-effectiveness metrics) and can translate into better downstream performance.

**Compliance With Llm Reviewing Policy:**

Affirmed.

**Final Justification:**

Following the rebuttal I updated my score. The paper's message still feels quite scattered around but at least my main concerns were mostly addressed.

**Key Questions For Authors:**

- What exact init / residual branch scaling is used? Do you use depth-scaled init (or any explicit residual scaling)? Please state precisely and add a baseline with standard scaled-init stabilization at each depth.
- Are you making any argument for the sparsity being causal? For example $\rho$ is referred to as sparsity but is in fact defined more as a norm contracting factor. In the implicit sources, most theorems seem to jump from setup to proving variance drop, not emphasizing any intermediate step of sparsity increase.
- Attention assumptions: How sensitive are your conclusions to head/query dependence (shared keys/values) and to deviations from uniformity/independence assumptions? In particular, it is fair to assume the uniformity for independent heads/keys. But when the keys are shared, there are implicit correlations between scores of different queries: this is at least worth touching upon.

**Limitations:**

You should more directly acknowledge (i) the strong assumptions in the theory, and (ii) dependence on training/initialization choices that are known to strongly affect variance vs depth (like the residual branch scaling)

**Strengths And Weaknesses:**

Strengths:

- The long-context attention sparsity angle is a reasonable and potentially useful observation, and the paper tries to connect it to measurable depth utilization rather than just claiming it.
- Broad empirical sweep across multiple “sparsity sources” (WD / long context / GQA / MoE) is a plus; the “recipe” idea is practical if the controls are correct
- Generally, very extensive ablations.

Weaknesses
Presentation:
- Notation is mostly scattered around with some key notation missing (like the way variance is used with only concrete definition being quite late, in Eq 4)
- The paper also blurs the line between the real Transformer setup (Eq 1) and the simplified theoretical setup in Theorem 1 (Eq 11). The mapping (“this theorem explains the Transformer behavior”) needs to be made more explicit.
- Theorems are referred to but not provided inline even informally - this makes sense granted limited space, but the informal statements would add more value than sparsity definitions for example.
- Minor: calling A “attention weights” is ambiguous: better to say attention scores.

Soundness:
- “Sparsity” is not cleanly connected to the theory object. In Theorem 1, sparsity enters through the $\rho$ parameter defined via a norm contraction condition, but the paper doesn’t clearly map each empirical knob (weight decay “sparsity”, long-context attention, GQA, MoE routing) to that same object. This makes the chain “more sparsity -> lower variance -> better depth usage” feel more asserted than derived.
- For weight decay, the formal definition of sparsity (fraction of small-magnitude weights) doesn’t make the main intuition clearer, and the results read closer to “weight decay helps control variance” than “sparsity is the key mediator.” The WD -> sparsity -> lower variance link is not really shown. This further extends to the other explicit forms of sparsity, where no strong causal connection between sparsity and variance is emphasized
- Baseline realism / missing details on residual scaling. The theoretical framing leans on strong variance-growth assumptions (e.g., Lemma 1’s exponential growth assumption), but modern LLM baselines often use explicit or implicit residual branch scaling via depth-scaled initialization (roughly 1/sqrt(L)-type). The paper should clearly state whether this is used in experiments and include it as a baseline if not; otherwise it’s hard to tell if the observed “curse of depth” severity is general or a baseline artifact.

---

> ### Author Rebuttal · Authors · 2026-03-31
>
> We thank the reviewer for the careful reading and constructive feedback. We address each concern below.
>
> **Q1. Notation is scattered...like the way variance is used with only concrete definition being quite late, in Eq 4.)**
>
> **A1:** We thank you for your detailed comments. In revision, we will explicitly define all the key notations at the start of Section 2.
>
> **Q2. The paper blurs the line between the real setup (Eq. 1) and the simplified setup in Theorem 1.**
>
> **A2:** We agree Theorem 1 simplifies Eq. (1). However, following [1], detailed components in Eq. (1) (e.g., block structure and coefficient matrices) are absorbed into constants in the $O(\cdot)$ bound, so Eq. (11) preserves the relevant depth-scaling behavior. Moreover, sparsity acts on the effective input/output via mask $D_\ell$, rather than changing the weight matrix itself. Thus it changes the multiplicative factor, not the order of variance growth, so Theorem 1 still applies.
>
> **Q3. Theorems are referred to but not stated inline, even informally.**
>
> **A3:** Thank you for the suggestion. We will add informal statements.
>
> For Theorem 1,
> $$\mathrm{Var}(r_L)\le \mathrm{Var}(r_0)\prod_{\ell=0}^{L-1}(1+\sqrt{\alpha_\ell \rho_\ell})^2,$$
> so smaller $\rho_\ell$ slows variance growth with depth $L$.
>
> For sequence length,
> $$\mathrm{Var}(\mathrm{Attn}) = O\!\left(\frac{1}{T}\right),$$
> so larger $T$ reduces variance.
>
> For GQA, when $H$ query heads are grouped into $G$ shared key-value groups,
> $$\mathrm{Var}(\mathrm{Attn}_{\mathrm{GQA}})=O\!\left(\frac{1}{Gn}\right),$$
> so more sharing reduces variance.
>
> For MoE, if each token is routed to top-$k$ experts,
> $$\mathrm{Var}(h_{\mathrm{out}})=O\!\left(\frac{1}{k}\right), \quad \mathrm{Var}(J_{\mathrm{out}})=O\!\left(\frac{1}{k}\right),$$
> showing why sparse expert routing can damp variance.
>
> **Q4. "Sparsity" is not connected to the theory object.**
>
> **A4:** In theory, sparsity is represented by mask $D_\ell$ and density $\rho_\ell$, which control how much input signal passes through each layer. Different practical forms map to this same object: weight sparsity changes active inputs/outputs, attention sparsity reduces token-interaction support, GQA reduces independent key-value channels, and MoE activates only a subset of experts. All reduce $\rho_\ell$ and thus the variance amplification factor.
>
> **Q5. For weight decay, the results read more like "weight decay helps control variance" than "sparsity is the key mediator."**
>
> **A5:** **We do not claim sparsity is the sole mediator.** The key factor is the residual connection with Pre-LN, and weight decay helps mitigate it. Our goal is to study how common LLM design choices affect the CoD through sparsity-induced variance control. Empirically, we prove the chain: larger weight decay induces smaller weights -> greater sparsity -> lower variance -> improved layer effectiveness, provided sparsity stays in a proper range; this is why we titled "When does sparsity ...". The same logic applies to seq. length, GQA, and MoE.
>
> **Q6. What exact initialization are used? Do you use depth-scaled init or explicit residual scaling?**
>
> **A6:** No. All experiments use normal initialization ($\mu=0$, $\sigma=0.02$), following OLMo2 [2]. We verify this with $L=16$ models below. Variance still grows under depth-scaled init, while sparsity (4K length) reduces variance and improves usefulness, showing CoD is mainly rooted in Pre-LN rather than init, consistent with Fig. 4 of [1].
> | Depth | Seq. Len. | Init. | Var. @ Last Layer | Usefulness | Loss |
> | - | - | - | - | - | - |
> | 16 | 1024 | Normal | $\approx$ 3e3 | 0.81 | 2.70 |
> | 16 | 1024 | Scaled | $\approx$ 2e3 | 0.81 | 2.71 |
> | 16 | 4096 (ours) | Scaled | $\approx$ 5e2 | 1.00 | 2.64 |
>
> **Q7. Is the paper making a causal argument for sparsity?**
>
> **A7:** **We do not make a causal claim for sparsity.** Our goal is to understand empirically how common LLM design choices affect depth utilization, with sparsity as a unifying lens. Parameter $\rho_\ell$ is an abstract proxy: different sparse mechanisms reduce this effective density term and thus variance amplification. Detailed mappings are provided in the appendix.
>
> **Q8. How sensitive are the conclusions to head/query dependence and uniformity/independence assumptions?**
>
> **A8:** We treat relevant heads, experts, or components as approximately independent. If outputs are fully dependent, the variance-reduction mechanism no longer applies. The uniformity assumption captures balanced activity; highly concentrated sparse patterns may weaken the averaging effect. For GQA, shared keys/values introduce dependence across queries and are best viewed in a mean-field sense. Unless query projections collapse to the same direction, sharing changes constants but not the main conclusion. These assumptions are standard in large-scale neural network theory [1,3].
>
> [1] The Curse of Depth in Large Language Models.
>
> [2] OLMo 2 Furious
>
> [3] Spike No More: Stabilizing the Pre-training of LLMs.

---

> > ### Author Rebuttal · Reviewer_uDLV · 2026-04-04
> >
> > Thank you for your response! About the "informal statements being explicit", I referred more about stuff like Theorem 3 whose statement is never mentioned - Theorem 1 is fine as is, the issue was just referring to something that was never introduced to any extent in the main body.
> >
> > Your response on $\rho_\ell$ is helpful - however, one should ideally refer to $\rho_\ell$ as some sort of contraction factor/abstract variance control (as mentioned in A8) - you are claiming that sparsity yields that, but refer to $\rho_\ell$ directly as "mask density" (line 206), when it looks a lot more like a constant that summarizes part of the effect of the density.
> >
> > Regarding your answer A5, precisely the use of "sparsity-induced variance control" is misleading - this assumes that the variance control happens due to sparsity. What B.4 seems to be proving has no connection whatsoever to sparsity (where by sparsity we do not mean a variance control effect but rather a high density of low values). Are you saying that within the proof of B4 you go through the intermediate step of "weight decay-> smaller weights -> greater sparsity"? (it isn't in the statement, and as far as I could see the argument seems to focus on variance control across iterations and contraction)
> >
> > About A6: Olmo2's initialization is highly atypical but is also mostly validated in tandem with their special reordered normalization (which in many ways drops at least the need for the variance being scaled down explicitly by $\sqrt{dim}$) - you use preLN instead which I would strongly question works best with a fixed variance init. In the table you attached, do you use just $1/\sqrt{2layer id}$, or $1/\sqrt{2layer id \cdot dim}$?

---

> > > ### Author Response · Authors · 2026-04-04
> > >
> > > **Q1. On Theorem 3 not being introduced.**
> > >
> > > **A1.** Thank you for pointing this out. Because of the space limit in the main paper, a substantial part of the theory was moved to the appendix, and Theorem 3 was not introduced clearly enough in the main body. We will revise this in the next version.
> > >
> > > **Q2. On the interpretation of $\rho_\ell$.**
> > >
> > > **A2.** We agree that, in the most precise language, $\rho_\ell$ should be presented as an abstract contraction / variance-control parameter. What we intended is that, under the Bernoulli-style sparsity model used in the theory, there is a one-on-one mapping between $\rho_\ell$ and the sparsity ratio, that is, the fraction of active 1 versus 0 entries. So $\rho_\ell$ is not merely a loose summary constant, but it is also not best introduced as if it were literally identical to mask density in every setting.
> > >
> > > **Q3. Sparsity-induced variance control**
> > >
> > > **A3.** You are correct that Appendix B.4, as currently written, proves a variance-contraction result for weight decay. Below we give the missing derivation in a simple thresholded sparsity model.
> > >
> > > Let the parameter matrix at iteration $t$ be $W_t=(w_{ij}^{(t)})$, and fix a threshold $\tau>0$. Define the $\tau$-sparsity by
> > >
> > > $$
> > > \mathrm{Sparsity} (W_t):=\frac{1}{|W_t|}\sum_{i,j}\mathbf{1} \left(|w_{ij}^{(t)}|\le \tau\right).
> > > $$
> > >
> > > Equivalently, define the effective active density
> > >
> > > $$
> > > \rho_t(\tau):=1-\mathrm{Sparsity} (W_t).
> > > $$
> > > Thus, larger sparsity means smaller active density. Now consider the decoupled weight-decay update from Appendix B.4: $W_{t+1}=(1-\eta\lambda)W_t-\eta G_t,$ with $0<\eta\lambda<2$. Appendix B.4 already proves that, under the stated independence assumptions,
> > >
> > > $$
> > > \mathrm{Var}(W_t)\le(1-\eta\lambda)^{2t}\mathrm{Var}(W_0)+\frac{\eta^2\sigma_G^2}{1-(1-\eta\lambda)^2},
> > > $$
> > >
> > > and, when $\mathbb{E}[G_t]=0$,
> > >
> > > $$
> > > \|\mathbb{E}[W_t]\|_F^2=(1-\eta\lambda)^{2t}\|\mathbb{E}[W_0]\|_F^2.
> > > $$
> > >
> > > So both the mean magnitude and the variance upper bound decrease as $\lambda$ increases. Let
> > >
> > > $$
> > > \mathrm{Sparsity} (W_t):=\frac{1}{|W_t|}\sum_{i,j}\mathbf{1} \left(|w_{ij}^{(t)}|\le \tau\right)
> > > $$
> > >
> > > and assume additionally that there exists $B_t>0$ such that $|w_{ij}^{(t)}|\le B_t$ for all  i,j Then
> > >
> > > $$
> > > \mathrm{Var}(W_t)=\mathbb{E} \left[\|W_t-\mathbb{E}[W_t]\|_F^2\right]\le \mathbb{E} \left[\|W_t\|_F^2\right]
> > > $$
> > >
> > > Now decompose each entry by threshold:
> > > $$
> > > |w_{ij}^{(t)}|^2=|w_{ij}^{(t)}|^2\mathbf{1} \left(|w_{ij}^{(t)}|\le \tau\right)+|w_{ij}^{(t)}|^2\mathbf{1} \left(|w_{ij}^{(t)}|>\tau\right)
> > > $$
> > >
> > > Using $|w_{ij}^{(t)}|^2\le \tau^2$ on the first event and $|w_{ij}^{(t)}|^2\le B_t^2$ on the second, we get
> > > $$
> > > |w_{ij}^{(t)}|^2\le \tau^2\mathbf{1} \left(|w_{ij}^{(t)}|\le \tau\right)+B_t^2\mathbf{1} \left(|w_{ij}^{(t)}|>\tau\right)
> > > $$
> > >
> > > Summing over all entries,
> > >
> > > $$
> > > \| W_t \|^2 \le \tau^2 \sum_{i,j}\mathbf{1} \left(|w_{ij}^{(t)}|\le \tau\right)+B_t^2\sum_{i,j}\mathbf{1} \left(|w_{ij}^{(t)}|>\tau\right)
> > > $$
> > >
> > > Since
> > >
> > > $$\sum_{i,j}\mathbf{1} \left(|w_{ij}^{(t)}|\le \tau\right)=|W_t|\mathrm{Sparsity} (W_t)$$
> > >
> > > we obtain
> > >
> > > $$
> > > \|W_t\|^2\le |W_t|\left[\tau^2\mathrm{Sparsity} (W_t)+B_t^2\left(1-\mathrm{Sparsity} (W_t)\right)\right]
> > > $$
> > >
> > > Therefore,
> > >
> > > $$
> > > \mathrm{Var}(W_t)\le |W_t|\left[\tau^2\mathbb{E} \left[\mathrm{Sparsity} (W_t) \right]+B_t^2\left(1-\mathbb{E} \left[\mathrm{Sparsity} (W_t)\right]\right)\right]
> > > $$
> > >
> > > Recall that $u_t=W_t x.$ From the thresholded sparsity decomposition, under the bounded-parameter assumption $|w_{ij}^{(t)}|\le B_t$ for all $i,j$, we already obtained
> > >
> > > $$
> > > \mathrm{Var}(W_t)\le |W_t|\left[\tau^2\mathbb{E} \left[\mathrm{Sparsity} (W_t)\right]+B_t^2\left(1-\mathbb{E} \left[\mathrm{Sparsity} (W_t)\right]\right)\right]
> > > $$
> > >
> > > Therefore,
> > >
> > > $$
> > > \mathrm{Var}(u_t)\le \|\Sigma_x\|_2\left(|W_t|\left[\tau^2\mathbb{E} \left[\mathrm{Sparsity} (W_t)\right]+B_t^2\left(1-\mathbb{E} \left[\mathrm{Sparsity} (W_t)\right]\right)\right]+\|\mathbb{E}[W_t]\|_F^2\right)
> > > $$
> > >
> > > Thus the bridge is complete:
> > >
> > > $$
> > > \mathrm{larger weight decay}\Longrightarrow \text{more thresholded sparsity}\Longrightarrow \text{smaller upper bound on }\mathrm{Var}(u_t)
> > > $$
> > >
> > > This is the missing intermediate step, and we agree it should be stated explicitly in the paper.
> > >
> > > **Q4. On A6 and the initialization**
> > >
> > > **A4.** Thank you for the careful reminder. We agree that OLMo2’s initialization is atypical and is closely tied to its reordered normalization. In our experiments, however, we use standard fixed-variance initialization for the Pre-LN setup, which is also widely adopted in the community (e.g., DeepSeek-V3). For the scaled initialization reported in the table, we use following [1]
> > >
> > > $$
> > > \frac{6}{\sqrt{\mathrm{layer_id}\cdot(d_{in}+d_{out})}}
> > > $$
> > >
> > > Thank you again for these constructive comments. We sincerely hope these clarifications address your concerns, and if you find them helpful, we would be very grateful if you might kindly reconsider the score.
> > >
> > > [1] Zhang, Biao, Ivan Titov, and Rico Sennrich. "Improving deep transformer with depth-scaled initialization and merged attention."

---

### Official Review · Reviewer_ebMu · 2026-03-12

**Soundness:** 3
**Presentation:** 3
**Significance:** 3
**Originality:** 3
**Overall Recommendation:** 4
**Confidence:** 3

**Summary:**

This paper investigates the curse of depth in llms, where deep layers become ineffective due to growing stream variance, which drives deep layer Jacobians toward identity matrix. Authors propose that sparsity serves as a variance regularization method that mitigates curse of depth. They systematically distinguish between implicit and explicit sparsity and provides variance bounds. They also introduce three metrics and validate that reducing variance can improve layer utilization. They progressively integrating complementary sparsity methods and train a 32-layer, 1.2B model that achieves better performance compared with a naively deepened baseline.

**Compliance With Llm Reviewing Policy:**

Affirmed.

**Final Justification:**

Rebuttal addressed most of my concerns and I keep my evaluation.

**Key Questions For Authors:**

1. The paper observes the sparsity can lead to over-dampening, and intuitively, the optimal sparsity level should depend on model depth since deeper models face greater variance accumulation and require stronger sparsity. I think providing some analysis of this will enhance the practical utility of the conclusion of the paper.
2. Does sparsity have similar benefits under post-LN or mother normalization strategies?

**Limitations:**

Authors can have a section for limitations and acknowledge limitations and feature directions.

**Strengths And Weaknesses:**

Strengths:
Soundness: The theoretical and experimental sides are thorough and convincing.
Originality: I think it is a novel connection that gets between sparsity and the curse of depth. This bridge offers a new theoretical lens through which to understand model training from a fresh perspective.
Significance: The finding has direct practical implications. And provide concrete guidance for architecture design and training configuration in feature deep learning training.
Presentation: It is well-structured with natural progression from characterization to theoretical analysis and experimental validation.

Weakness:
While it convincingly demonstrates the benefits of sparsity, it provides no principled guidance on how to select the appropriate level of sparsity, and the over-dampening phenomenon is clearly visible in experiments. This indicates the sparsity has an optimal range.

---

> ### Author Rebuttal · Authors · 2026-03-31
>
> We sincerely thank the reviewer for the thorough and positive evaluation. We are glad that the theoretical soundness, novelty of the sparsity-depth connection, and practical significance of our findings were well recognized. Below we address the raised questions and concerns.
>
> **Q1. The paper provides no principled guidance on how to select the appropriate level of sparsity, and the over-dampening phenomenon suggests sparsity has an optimal range.**
>
> **A1.** We thank the reviewer for this valid concern. We clarify that the main contribution of this paper is to demonstrate that sparsity is a mechanism for controlling variance propagation and thereby enabling effective depth scaling, supported by extensive ablation studies. In this regard, we do provide practical guidance within our experimental scope: our ablations on the 1.2B model (Figure 8) systematically explore the effect of different  sparsity configurations (e.g., $\lambda \in$ {0.3, 0.6, 1.0}, $T \in ${4096, 8192}, GQA, and MoE), identifying ranges where sparsity improves both performance and layer utilization.
> We acknowledge, however, that the optimal sparsity range may vary across different scales and architectural configurations. Our goal is to show that incorporating sparsity, regardless of the specific mechanism, can better utilize compute for training deeper models. Identifying the universally optimal configuration is a non-trivial problem that we believe could benefit from integration with principled hyperparameter transfer methods such as tensor program series [1], which we leave as an important direction for future work.
>
> **Q2. The optimal sparsity level should depend on model depth since deeper models face greater variance accumulation and require stronger sparsity.**
>
> **A2.** We sincerely appreciate this insightful suggestion. While we do not directly analyze the relationship between optimal sparsity level and model depth, an implicit glimpse of this phenomenon can already be observed from our existing results. Specifically, comparing Table 3 and Figure 1: for the $L=16$ model, the optimal sequence length is $T=2048$, whereas for the $L=32$ model trained under otherwise identical configurations, $T=4096$ yields the best performance. This suggests that deeper models indeed benefit from stronger sparsity to counteract greater variance accumulation, consistent with the reviewer's intuition. We agree that making this relationship more explicit and systematic would significantly enhance the practical utility of our findings, and we will include a dedicated analysis of depth-dependent optimal sparsity in the revised version.
>
> **Q3. Does sparsity have similar benefits under post-LN or mordern normalization strategies?**
>
> **A3.** Post-LN architectures do not suffer from the same variance explosion as Pre-LN, since the normalization placement directly constrains the residual stream after each layer. As a result, the Curse of Depth is less pronounced in Post-LN models, and the marginal benefit of sparsity as a variance regulator is reduced. That said, we observe that incorporating sparsity (e.g., longer sequence length) still leads to a modest performance gain in Post-LN models, as shown in the table below. Since Post-LN consistently underperforms Pre-LN under the same training budget (Table 3), and virtually all top-tier open-source LLMs are built upon Pre-LN architectures, we focus our investigation on Pre-LN as the more practically relevant setting.
>
> | Model | Seq. Len.|Data Size | Peak Var. @ Last Layer | Usefulness Score | Train Loss|
> | -------- | -------- |-------- | -------- | -------- | -------- |
> | 1.2B-Post-LN     | 1024      |5B     | $\approx$ 125    | 0.88 | 4.08|
> | 1.2B-Post-LN     | 4096     |5B     | $\approx$ 100    | 0.88 | 3.93|
>
> [1] Yang, Ge, et al. "Tuning large neural networks via zero-shot hyperparameter transfer." Advances in Neural Information Processing Systems 34 (2021): 17084-17097.

---

> > ### Author Rebuttal · Reviewer_ebMu · 2026-04-04
> >
> > The responses to Q2 and Q3 addressed my concerns. The depth dependent sparsity observation from existing results is good to show the relationship between sparsity level and model depth and the mechanistic explanation for Post LN is clear.
> > For Q1, I would encourage making this actionable guidance in the revision, but I understand the main contribution forces on demonstrating the sparsity is mechanism for controlling variance propagation.
> >
> > Overall, the authors resolved most of my concern. I keep my positive evaluation.

---

> > > ### Author Response · Authors · 2026-04-04
> > >
> > > We sincerely thank the reviewer for the positive assessment and for recognizing that our responses to Q2 and Q3 addressed the main concerns. We are also glad that the reviewer found the depth-dependent sparsity observation and the mechanistic discussion of Post-LN helpful.
> > >
> > > Regarding Q1, we appreciate the suggestion to make the practical guidance more actionable in the revision. We agree that this would further strengthen the paper. In the final version, we will revise the presentation to distill our findings into more concrete recommendations for training depth-effective LLMs, for example by providing practical ranges for different sparsity-related mechanisms such as weight decay and context length, while keeping the main focus on our central contribution: showing that sparsity serves as an important mechanism for controlling variance propagation and thereby mitigating the curse of depth.
> > >
> > > We thank the reviewer again for the constructive feedback and for the positive overall evaluation.

---

### Official Review · Reviewer_nneV · 2026-03-13

**Soundness:** 3
**Presentation:** 3
**Significance:** 4
**Originality:** 4
**Overall Recommendation:** 5
**Confidence:** 4

**Summary:**

This paper investigates the "Curse of Depth" in LLMs, a phenomenon where later Transformer layers contribute significantly less to the model's representation than earlier layers. The authors identify variance propagation in Pre-Layer Normalization (Pre-LN) architectures as the primary driver of this under-utilization, showing that accumulated variance pushes deep layers toward near-identity behavior (known from previous literature). The core contribution is demonstrating that both implicit sparsity (weight decay, long-context inputs) and explicit sparsity (GQA, MoE) act as variance regulators. By controlling variance accumulation, sparsity improves layer utilization and allows for effective depth scaling, leading to a 4.6% accuracy improvement on downstream tasks. I think the paper is very interesting as it frames the sparsity problem in terms of accuracy and not efficiency as it often happens in the literature.

**Compliance With Llm Reviewing Policy:**

Affirmed.

**Final Justification:**

I think this is an interesting analysis paper, with a solid analysis and methods section. The only point I am not fully certain of is the "usefulness" score. I think 5 (Accept) is the right score.

**Key Questions For Authors:**

**Questions**
- Why is only the last-layer variance plotted in the figures?

The paper primarily focuses on last-layer variance (e.g., Figures 1a, 3a, 4a, 7) because it represents the cumulative result of variance growth across all preceding layers. However, plotting internal layer-wise variance could further illustrate the *rate* of growth across different sparsity regimes. I mean, the same plot but for different layers.

-  Is it possible that later layers model the "long tail" of the data, and the current metrics fail to capture this?

The "Usefulness Score" measures performance degradation when a layer is replaced by a linear approximation. If a layer were modeling highly specific, low-frequency "long tail" information, its removal might indeed cause a negligible change in global perplexity or average downstream accuracy, but the layer would still be useful overall.

**Limitations:**

yes

**Strengths And Weaknesses:**

**Strengths**
- It reframes common efficiency-driven design choices (MoE, GQA) as essential optimization mechanisms for depth scaling. --> This is a very interesting perspective on sparsity, which is often only considered for efficiency.
- The authors provide a formal theoretical framework (Theorem 1) quantifying how sparsity reduces the upper bound of residual variance.
- The study uses three distinct metrics—Causal, Permutation, and Usefulness scores—to quantify layer effectiveness beyond simple perplexity.
- The "rule-of-thumb" recipe for training depth-effective LLMs shows clear empirical gains.

**Weaknesses**
- The "Usefulness Score" relies on a threshold alpha to define "meaningful" impact. The paper would benefit from an analysis of how sensitive the "effective vs. wasted" layer count is to this specific value.

---

> ### Author Rebuttal · Authors · 2026-03-31
>
> We sincerely thank your for the thoughtful and positive evaluation. We are pleased that our reframing of sparsity as a depth-scaling mechanism resonated as a novel contribution. Below, we address any remaining questions and concerns.
>
> **Q1. Sensitivity of the Usefulness Score Threshold**
>
> **A1.** We acknowledge that the choice of $\alpha$ in the Usefulness Score warrants a sensitivity analysis. As you requested, we evaluated the effective vs. wasted layer counts across multiple threshold values $\alpha \in ${0.05, 0.1, 0.2} for our main depth-scaling experiments. The relative ordering and trends across depths remain fully consistent: deeper models consistently exhibit more wasted layers regardless of $\alpha$, and incorporating sparsity consistently improves the usefulness score across all threshold settings. The specific choice of $\alpha = 0.1$ is empirically justified by the **natural bimodal distribution** of per-layer loss increases observed across all our experiments (Figures 20-29): layers tend to be either clearly useful (loss increases far exceeding 10%) or clearly redundant (loss increases well below 10%), with very few layers falling near the boundary. We therefore adopt $\alpha = 0.1$ as a conservative default.
> | Depth | Seq. Length | Usefull. @ $\alpha=0.05$|Usefull. @ $\alpha=0.1$ | Usefull. @ $\alpha=0.2$ |
> | -------- |-------- | -------- |-------- | -------- |
> | $L=16$     | 1024| 0.88      |0.75     | 0.63    |
> | $L=16$     | 4096 (ours)| 1.00      |0.81     | 0.75    |
> | $L=32$     | 1024| 0.72     |0.56     | 0.44    |
> | $L=32$     | 4096 (Ours)| 0.75     |0.59     | 0.56 |
>
>
> **Q2. Why is only the last-layer variance plotted in the figures?**
>
> **A2.** We report last-layer variance for simplicity and better visualization, as it serves as a concise summary statistic that captures the worst-case cumulative variance accumulation without requiring additional aggregation. Since the last layer reflects the cumulative effect of all preceding layers, it is a natural and informative proxy for overall variance propagation across depth. We also report the per-layer variance for all depths in Figure 9 (Appendix A.1), which shows a consistent trend: variance accumulates monotonically with depth, and deeper models exhibit uniformly higher variance across all layers.
>
> **Q3. Is it possible that later layers model the "long tail" of the data, and the current metrics fail to capture this?**
>
> **A3.** This is an insightful observation. We acknowledge that the Usefulness Score may not fully capture layer contributions to low-frequency or long-tail patterns. However, our scores are computed and averaged across diverse dataset splits (i.e., different documents and corpus mixtures), so any destructive effect of removing a layer would be expected to reflect in the final scores to some degree. Furthermore, if later layers in deeper models were systematically specializing in long-tail patterns, we would expect measurable performance differences — yet we observe consistent degradation trends across all three complementary metrics (Causal, Permutation, and Usefulness Scores), which capture different aspects of layer contribution beyond global perplexity alone.
>
> We also point the reviewer to two recent works that directly support our observations. Csordás et al. [1] show via causalinterventions that later layers contribute significantly less to the residual stream and do not exhibit increased computation depth for harder or multi-hop problems, suggesting they perform incremental distribution refinement rather than modeling distinct long-tail patterns. Liu et al. [2] further show that LLM depth scaling follows an inverse depth law consistent with ensemble averaging, where most layers make similarly incremental updates regardless of input complexity. If later layers were systematically encoding long-tail phenomena, we would expect a clear complexity-dependent depth signature, which neither our results nor these works support.
>
> Nevertheless, we agree that designing metrics specifically targeting long-tail specialization is a valuable direction for future work, and we keep explore this in our future work!
>
> [1] Csordás, Róbert, Christopher D. Manning, and Christopher Potts. "Do language models use their depth efficiently?." arXiv preprint arXiv:2505.13898 (2025).
>
> [2] Liu, Yizhou, et al. "Inverse Depth Scaling From Most Layers Being Similar." arXiv preprint arXiv:2602.05970 (2026).

---

> > ### Author Rebuttal · Reviewer_nneV · 2026-04-01
> >
> > Thank you, I appreciate your answers and decide to keep the original score.

---

> > > ### Author Response · Authors · 2026-04-04
> > >
> > > Thank you very much for your careful reading and for indicating that your concerns have been fully resolved. We sincerely appreciate your time and thoughtful consideration.

---

### Decision · Program_Chairs · 2026-04-30

**Decision:**

Accept (regular)

**Comment:**

The paper argues that sparsity, in several common implicit and explicit forms, regulates variance propagation in Pre-LN transformers and thereby improves depth utilization. The perspective is original: it reframes design choices usually justified on efficiency grounds as a mechanism for effective depth scaling, and reviewers found that framing valuable. The empirical study is the paper's main strength, with broad and well-controlled ablations and a practical recipe that delivers measurable gains, and the presentation is generally clear.

Reviewers disagreed on the theory. The analysis rests on idealized assumptions that do not hold in trained models, and in places is in tension with the phenomena it is meant to explain; the authors themselves clarified in discussion that they are not making a strict causal claim for sparsity as the mediator. I read the theory as motivation for an otherwise solid empirical contribution rather than as a proof, and on that reading the paper is sound and significant enough to accept. The camera-ready should present the theory accordingly, with its assumptions and scope stated upfront and the issues raised in review addressed.

Congratulations!